# Exchange of $CO_2$ in Arctic tundra: impacts of meteorological variations and biological disturbance

Efrén López-Blanco[1,2], Magnus Lund[1], Mathew Williams[2], Mikkel P. Tamstorf[1], Andreas Westergaard-Nielsen[3], Jean-François Exbrayat[2,4], Birger U. Hansen[3], Torben R. Christensen[1,5]

[1] Department of Biosciences, Arctic Research Center, Aarhus University, Frederiksborgvej 399, 4000 Roskilde, Denmark
[2] School of GeoSciences, University of Edinburgh, Edinburgh, EH93FF, UK
[3] Center for Permafrost (CENPERM), Department of Geosciences and Natural Resource Management, University of Copenhagen, Oester Voldgade 10, 1350 Copenhagen K, Denmark
[4] National Centre for Earth Observation, University of Edinburgh, Edinburgh, EH93FF, UK

[5] Department of Physical Geography and Ecosystem Science, Lund University, Sölvegatan 12, 223 62 Lund, Sweden

*Correspondence to*: Efrén López Blanco (elb@bios.au.dk)

**Keywords:** Arctic tundra, Greenland, Atmospheric $CO_2$, Net Ecosystem Exchange, Gross primary production, Ecological Respiration, meteorological responses, insect outbreak.

**Abstract.** An improvement in our process-based understanding of carbon (C) exchange in the Arctic, and its climate sensitivity, is critically needed for understanding the response of tundra ecosystems to a changing climate. In this context, we analyzed the net ecosystem exchange (NEE) of $CO_2$ in West Greenland tundra (64° N) across eight snow-free periods in eight consecutive years, and characterized the key processes of net ecosystem exchange, and its two main modulating components: gross primary production (GPP) and ecosystem respiration ($R_{eco}$). Overall, the ecosystem acted as a consistent sink of $CO_2$, accumulating -30 g C $m^{-2}$ on average (range -17 to -41 g C $m^{-2}$) during the years 2008-2015, except 2011 (source of 41 g C m-2) that was associated with a major pest outbreak. The results do not reveal a marked meteorological effect on the net $CO_2$ uptake despite the high inter-annual variability in the timing of snowmelt and start and duration of the growing season. The ranges in annual GPP (-182 to -316 g C $m^{-2}$) and $R_{eco}$ (144 to 279 g C $m^{-2}$) were >5 fold larger than the range in NEE. Gross fluxes were also more variable (Coefficients of variation are 3.6 and 4.1 % respectively) than for NEE (0.7 %). GPP and $R_{eco}$ were sensitive to insolation and temperature; and there was a tendency towards larger GPP and $R_{eco}$ during warmer and wetter years. The relative lack of sensitivity of NEE to meteorology was a result of the correlated response of GPP and $R_{eco}$. During the snow-free season of the anomalous year of 2011, a biological disturbance related to a larvae outbreak reduced GPP more strongly than $R_{eco}$. With continued warming temperatures and longer growing seasons, tundra systems will increase rates of C cycling. However, shifts in sink strength will likely be triggered by factors such as biological disturbances, events that will challenge our forecasting of C states.

## 1 Introduction

Quantifying the climate sensitivity of carbon (C) stocks of the terrestrial biosphere is a major challenge for Earth system science (Williams et al., 2005). In the Arctic, organic soil C storage has the potential for very large C releases following thaw (Koven et al., 2011) that could create a positive feedback on climate change and accelerate the rate of global warming. Recent reviews have estimated the Arctic terrestrial C pool to be 1400-1850 Pg C, more than twice the size of the atmospheric C pool (Hugelius et al., 2014; McGuire et al., 2009; Tarnocai et al., 2009) and approximately 50% of the global soil organic C pool (AMAP, 2011; McGuire et al., 2009). Further, Arctic ecosystems have experienced an intensified warming tendency, reaching almost twice the global average (ACIA, 2005; AMAP, 2011; Callaghan et al., 2012c; Serreze and Barry, 2011). The projected Arctic warming is also expected to be more pronounced in coming years (AMAP, 2011; Callaghan et al., 2012a; Christensen et al., 2007; Grøndahl et al., 2008; Meltofte et al., 2008) and temperature, precipitation and growing season length will likely increase in the Arctic (ACIA, 2005; Christensen et al., 2007; Christensen et al., 2004; IPCC, 2007). Given this situation, an improvement in our process-based understanding of $CO_2$ exchanges in the Arctic, and their climate sensitivity, is critical (McGuire et al., 2009).

Measuring the inter-annual C exchange variability in the Arctic tundra is challenging due to extreme conditions and the patchy nature of the landscape linked to micro-topography. Different eco-types are linked to different C exchange rates (Bubier et al., 2003). Synthesis studies have found a significant spatial variability in NEE (Lafleur et al., 2012; Mbufong et al., 2014) between different tundra sites (Lindroth et al., 2007; Lund et al., 2010) and also large temporal variability within sites (Aurela et al., 2004; Aurela et al., 2007; Christensen et al., 2012; Grøndahl et al., 2008; Lafleur et al., 2012). Minor variations in the key process of photosynthesis (gross primary production, GPP) and ecosystem respiration ($R_{eco}$) may promote important changes in the sign and magnitude of the C balance (Arndal et al., 2009; Elberling et al., 2008; IPCC, 2007; Lund et al., 2010; Tagesson et al., 2012; Williams et al., 2000). With continued warming temperature and longer growing seasons, tundra systems will likely have enhanced GPP and $R_{eco}$ rates, but long-term data to investigate and quantify these responses is rare. Further, the effects on net $CO_2$ sequestration are not known, and may be altered by long-term processes such as vegetation shifts and short-term disturbances like insect pest outbreaks, complicating the prognostic forecast of upcoming C states (Callaghan et al., 2012b; McGuire et al., 2012). Consequently, there is a need to understand how C cycle behaves over time scales from days to years, and the links to environmental drivers. There is a lack of reference sites in the Arctic from where full measurement-based data are available, documenting carbon fluxes at the terrestrial catchment scales. Here we investigate the functional responses of C exchange to environmental characteristics across eight snow-free periods in eight consecutive years in West Greenland.

In recent decades, eddy covariance has become a fundamental method for carbon flux measurements at the landscape scale (Lasslop et al., 2012; Lund et al., 2012; Reichstein et al., 2005). Eddy covariance measurements of land-atmosphere fluxes, or Net Ecosystem Exchange (NEE), of $CO_2$ can be gap-filled and subsequently separated into the modulating components of GPP and $R_{eco}$ using flux partitioning algorithms (Reichstein et al., 2005). These techniques are critical for providing a better understanding of the C uptake versus C release behaviour (Lund et al., 2010); but they also allow for an examination of the environmental effects on ecological processes (Hanis et al., 2015). However, large gaps in the measured fluxes may introduce significant uncertainties in the C budget estimations. Moreover, GPP and $R_{eco}$ estimates can be calculated in different ways. Some algorithms fit an instantaneous temperature-respiration curve to night-time data to calculate $R_{eco}$ and estimate GPP (Lasslop et al., 2012; Reichstein et al., 2005); others calculate $R_{eco}$ from a light-response curve (Gilmanov et al., 2003; Lindroth et al., 2007; Lund et al., 2012; Mbufong et al., 2014; Runkle et al., 2013). Unfortunately, different interpretations of the flux gap-filling and partitioning lead to different estimates of NEE, GPP and $R_{eco}$, as well as undefined uncertainties.

The main objectives of this paper are (1) to explore the uncertainties in NEE gap-filling and partitioning obtained from different approaches, (2) to determine how C uptake and C storage respond to the meteorological variability, and (3) to

identify how the environmental forcing affects not only the inter-annual variability, but also the hourly, daily, weekly and monthly variability of NEE, GPP and $R_{eco}$. The intention of this paper is to elaborate on the information gathered in an existing catchment area under an extensive cross-disciplinary ecological monitoring program in low Arctic West Greenland, established under the auspices of the Greenland Ecosystem Monitoring (GEM) (http://www.g-e-m.dk). Using a long-term (8 years) dataset to explore uncertainties in NEE gap-filling and partitioning methods and to characterise the inter-annual variability of C exchange in relation to driving factors can provide a novel input into our understanding of land-atmosphere $CO_2$ exchange in Arctic regions. Our overarching hypothesis was that both GPP and $R_{eco}$ would respond positively to warmer and longer growing seasons; but, that NEE response to warming would be more complex and variable (positive or negative), depending on subtle balances between plant and microbial climate sensitivity.

## 2 Materials and methods

### 2.1 Site description

Field measurements were conducted in the low Arctic Kobbefjord drainage basin, South-western Greenland (64° 07' N; 51° 21' W) (Figure 1a). The study area is located ~20 km SE of Nuuk, the Greenlandic capital. Kobbefjord has been subject to extensive environmental research activities (the Nuuk Ecological Research Operations) since 2007 (http://www.nuuk-basic.dk). The lowland site is located 500 meters from the South-eastern shore of the bottom of Kangerluarsunnguaq Fjord (Kobbefjord), and 500 meters from the Western shore of the 0.7 $km^2$ lake called "Badesø" (Figure 1b). Three glaciated mountains, all above 1000 m. asl., surround the site. The landscape consists on a fen area surrounded by heath, copse and bedrock. The current fen vegetation is dominated by *Scirpus caespitosus*, whereas the surroundings are dominated by heath species such as *Empetrum nigrum*, *Vaccinium uliginosum*, *Salix glauca* and copse species such as *S. glauca* and *Eriophorum angustifolium* (Bay et al., 2008). Kobbefjord belongs to the "Arctic Shrub Tundra" (bioclimate zone E) according to The Circumpolar Arctic Vegetation Map (CAVM Team, 2003; Walker et al., 2005). This map is based on the summer warmth index (SWI), which is the sum of the monthly mean temperature above 0 °C from May to September and the southernmost bioclimatic zone E has the limits 20-35. In 2010 and 2012, the weather conditions led the area to experience temperatures from warmer climatic zones (SWI ca. 36 and 35 respectively). For the 1961-1990 period, the mean annual air temperature was -1.4 °C and the annual precipitation was 750 mm (Cappelen, 2013). The sun light hours between May and September range from 14 to 21 hours. Outcalt's frost number (Nelson and Outcalt, 1987) indicates that discontinuous permafrost should be present, although no permafrost has been found. Nonetheless, thin lenses of ice may remain until late summer.

### 2.2 Measurements

We have used eddy covariance (EC) data on NEE, measured during the snow-free period from 2008 to 2015. Measurements typically started around the end of the snowmelt (ca. May-June) and extended until the freeze-in period (between September-October). Once the snow melts, the growing season (i.e. the part of the year when the weather conditions allow plant growth) has been reported as the most relevant period defining both spatial (Lund et al., 2010; Mbufong et al., 2014) and temporal (Aurela et al., 2004; Groendahl et al., 2007; Lund et al., 2012) $CO_2$ variability. The EC measurements were conducted in the EddyFen station (Figure 1b and 1c), located in a wet lowland, 40 m. asl. The EC tower is equipped with a closed-path infrared $CO_2$ and $H_2O$ gas analyzer LI-7000 (LI-COR Inc, USA) and a 3D sonic anemometer Gill R3-50 (Gill Instruments Ltd, UK). The anemometer was installed at a height of 2.2 m, while the air intake was attached 2.0 m above terrain on the steel stand. Adjacent to the EddyFen station, an independent system (Figure 1b and 1c) measures round-the-clock net $CO_2$ fluxes using an automatic chamber (AC) method based on Goulden and Crill (1997). The transparent chambers, each covering a known surface area of 60 cm by 60 cm, with a height of 30 cm, can be opened and closed by the computer in succession for 10 min every hour. When the chamber closes, a $CO_2$ analyzer (SBA-4, PP Systems, UK) monitors both the

$CO_2$ concentration by a close loop of tubing (further information about the set up can be found in Mastepanov et al. (2012). Nearly 20 m from the EddyFen station, the automated SoilFen (Figure 1b and 1c) station provides environmental variables such as air and surface temperature (Vaisala HMP45C), soil temperature at different depths (Campbell scientific 10ST) and relative humidity (Vaisala HMP45C). Two kilometres from these stations, an automatic weather station provides complementary ancillary data such as short & long wave radiation (with a CNR1 instrument), photosynthetic active radiation (with a Kipp & Zonen PAR Lite instrument), precipitation (using an Ott Pluvio instrument) and snow depth (with a Campbell Scientific SR 50). The water table depth data has been monitored using a piezometer located next to each of the six auto chambers. Finally, a robust daily estimate of the timing of snowmelt was analyzed at a pixel level from a time-lapse camera (HP e427) located at 500 m. asl. (Westergaard-Nielsen et al., 2013).

## 2.3 Data handling

### 2.3.1 Data collection and pre-processing

Data collection from the EddyFen station was performed using Edisol software (Moncrieff et al., 1997). Raw data files were processed using EdiRe software (version 1.5.0.32, R. Clement, University of Edinburgh) calculating the $CO_2$ fluxes on a half hourly basis. The flux processing integrated despiking (Højstrup, 1993), 2D rotation, time lag removal by covariance optimization, block averaging, frequency response correction (Moore, 1986) and Webb-Pearman-Leuning correction (Webb et al., 1980). For more information, see Westergaard-Nielsen et al. (2013). Ancillary data (air temperature, soil temperature, incoming short wave radiation, relative humidity, PAR and precipitation) have been temporally resampled using R (https://www.r-project.org/). Time-series-related packages such as *zoo* (Zeileis and Grothendieck, 2005), *xts* (Ryan and Ulrich, 2014) and *lubridate* (Grolemund and Wickham, 2011) were used to get the ancillary data aligned with the flux data in half-hourly basis.

### 2.3.2 Generating robust and complete flux time series

Before the $CO_2$ flux time series were analysed, we applied three different processing techniques (u*filtering, gap-filling and partitioning) to (1) filter the NEE data for quality, (2) fill the NEE gaps and (3) separate NEE into GPP and $R_{eco}$. The identification of periods with insufficient turbulence conditions (indicated by low friction velocity u*) is important to avoid biases and uncertainties in EC fluxes. To control the data quality, the u* thresholds were bootstrapped by identifying conditions with inadequate wind turbulence according to the method described in (Papale et al., 2006). We subsetted the data to similar environmental conditions, aside from friction velocity: 8 years and 7 temperature classes. Within each year/temperature subset the u* threshold (5%, 50% and 95% of bootstrap) was estimated in 1000 samples per year. We used the subsequent gap-filling and partitioning based on these different subsets to propagate the uncertainty of u* threshold estimation across NEE, GPP and $R_{eco}$.

Our gap-filling method was similar to Falge et al. (2001), using the marginal distribution sampling (MDS) algorithm, re-adapted from Reichstein et al. (2005) in REddyProc (Reichstein and Moffat, 2014). MDS takes into account similar meteorological data available with different window sizes (Moffat et al., 2007). Parallel to this approach, we also gap-filled the original EC NEE data with an independent AC NEE dataset (2010-2013). AC data were collected simultaneously with EC data, and so we can used them as a cross check. The EC NEE was predicted from AC NEE based on linear regression models. The subsequent product was gap-filled using the MDS algorithm (REddyProc).

We separated NEE into its two main components (GPP and $R_{eco}$) using two approaches: (1) the REddyProc partitioning tool (Reichstein and Moffat, 2014) and (2) a light response curve (LRC) approach (Lindroth et al., 2007; Lund et al., 2012). A brief description of each flux partitioning method is provided in the supplementary material (Equations S1). After the flux partitioning comparison, we used ReddyProc-based GPP and $R_{eco}$ estimates on further analyses.

### 2.3.3 Flux uncertainties

In order to estimate the NEE gap-filling uncertainty, we assessed three different sources of uncertainty. First, we addressed the 95% confidence interval of the EC prediction based on AC data. Second, we inferred the random uncertainty of filled half-hourly values by the spread of variable with otherwise very similar environmental conditions. REddyProc uses the gap-filling to estimate an observation uncertainty also for the measured NEE, by temporarily introducing artificial gaps (T. Wutzler and M. Migliavacca (BGC-Jena), personal communication). Finally, we assessed the effect of uncertainty in the estimate of the u* threshold. In the u*-NEE relationship we want to exclude the probably false low fluxes (absolute NEE values) at low u*. When choosing a lower u* threshold, the associated lower flux will contribute to the gap-filling and the annual sums. Therefore, there is a tendency of a lower absolute NEE associated with lower u*. The difference between the 5% and 95% of bootstrap provides a means of the uncertainties based on the u* filters. We summed and propagated all these sources of uncertainties over time. The GPP and $R_{eco}$ uncertainties include the bias from the one-to-one flux comparison obtained from each model. The micrometeorological sign convection used in this study present uptake fluxes (GPP) as negative, while the released fluxes ($R_{eco}$) are shown as positive.

### 2.4 Identifying environmental forcing

Snow and phenology related variables such as end of the snowmelt period and the start, end and length of the growing season are important components shaping the Arctic $CO_2$ dynamics. In this study we defined the end of the snowmelt period as the day of year when more than 80% of the surface of the fen was considered snow-free; the threshold was chosen in agreement with suggestions previously reported in Hinkler et al. (2002) and Westergaard-Nielsen et al. (2015). For the start, end and length of the growing season ($GS_{start}$, $GS_{end}$, $GS_{length}$); the $GS_{start}$ and the $GS_{end}$ were defined as the first and last day when the consecutive 3-day NEE average was negative (i.e. $CO_2$ uptake) and positive (i.e. $CO_2$ release) respectively (Aurela et al., 2004), while $GS_{length}$ is the number of days between $GS_{start}$ and $GS_{end}$).

A Random Forest machine-learning algorithm (Breiman, 2001; Pedregosa et al., 2011) was utilized in a data-mining exercise to identify how the environmental controls affect the variability of NEE, GPP and $R_{eco}$. Random forest calculates the relative importance of explanatory variables over the response variables. Here, we use photosynthetic active radiation (PAR), air temperature ($T_{air}$), precipitation (Prec) and vapor pressure deficit (VPD) to explain the response of C fluxes (NEE, GPP and $R_{eco}$) to climate variability. Each decision tree in the forest is trained on different random subset of the same training dataset. The Random Forest is a classifier that groups explanatory variables and, in each final cluster, a multiple linear regression is built to reproduce fluxes as function of driving factors. This approach has been used to extrapolate maps of biomass (Baccini et al., 2012; Exbrayat and Williams, 2015). This version of Random Forest sums the relative importance of each variable from 0% up to 100 %, which correspond to the fraction of decision in which a variable is involved to cluster the data. We applied Random Forest to assess the relative importance of PAR, $T_{air}$, Prec and VPD at different temporal scales (hourly, daily, weekly and monthly), aggregating them at the time scale indicated and lumping all the years together. (Table S1; supplementary material). Moreover, we also evaluated the diurnal, seasonal and annual pattern for each explanatory variable (data binned per hour, this is one Random Forest per hour of the day, day of the year and year respectively). To make sure that these results were not an artefact of the partitioning method that is based on a relationship between hourly $R_{eco}$ and $T_{air}$, we performed the same analyses using day-time and night-time only hourly NEE as respective proxies for GPP and $R_{eco}$. Based on these results (Table S2; supplementary material) we concluded that the approach was robust for the Kobbefjord site.

# 3 Results

## 3.1 Inter-annual and seasonal variation of environmental and phenological variables

The annual mean temperature documented from Nuuk (-0.5 °C) and Kobbefjord (-0.4 °C) in the 2008-2015 period were generally warmer compared to the long time series between 1866 and 2007 (Cappelen (2016); Figure S1; supplementary material), with an annual temperature average of -1.5 °C. The 2008-2015 period temperature also exhibited larger variability (Coefficients of variation (CV) = 283.3 %) compared to the 1866-2007 period (CV = 79.3 %). The 2008-2015 mean annual temperature measured in Kobbefjord fluctuated between -1.7 °C in 2011 and 3.4 °C in 2010. Moreover, the mean annual precipitation documented from the nearby station of Nuuk (885 mm) and the one measured across the eight years study in Kobbefjord (862 mm) were both significantly higher than the 1931-2007 mean (689 mm), although less variable (CV = 30.8 % and 24.5 % respectively). Overall, 2008, 2009, 2010, 2012, 2013 and 2014 have shown warmer and wetter anomalies while 2011 and 2015 presented colder and drier anomalies compared to the long-term mean (Figure 2a). Among the eight study years (figure 2b), the temperature and precipitation anomalies in the warm season (June to September) ranged from about -1°C (2011, 2013 and 2015) to +1.5°C (2010) and -96 mm (2011) to about +125 mm (2012 and 2013), respectively. The cold season (October to May) anomalies have shown greater variability compared to the warm season, and 2010, 2012 and 2013 experienced warmer and wetter winters, while 2011 and 2015 were colder and drier.

The end of the snowmelt period and the growing season start/length presented high inter-annual variability (CV were 9.5, 9.0 and 19.0 %, respectively). Kobbefjord became snow-free in DOY 154 (June 3[rd] for non-leap years, SD=15) on average. On average, the site switched from being a source of $CO_2$ to a sink ($GS_{start}$) on DOY 175 (June 24[th], SD=20), and remained so ($GS_{end}$) until DOY 241 (July 29[th], SD=8.4)(Table 1). The $GS_{start}$ and the $GS_{length}$ did not follow a consistent pattern among the analysed years, the growing season timing have fluctuated substantially. The high inter-annual variability of the $GS_{start}$ correlated with variations in temperature, end of snowmelt period and VPD ($p<0.05$). Highest variability was observed during 2009-2012. The 2010's $GS_{length}$ was nearly twice as long as to 2011. Indeed, $GS_{start}$ in 2011 differs only by 26 days with the $GS_{end}$ in 2010.

## 3.2 Data processing and quality

The NEE gap-filling and subsequent partitioning obtained from different approaches exposed inconsistencies in performance and specific uncertainties in the seasonal C budget calculation. During the eight study snow-free periods, data gaps made up 46.5 % of the record from the EddyFen station, due to unfavourable micro-meteorological conditions, instrument failures, maintenance and calibration (Jensen and Christensen 2014), but also due to the rejection of low quality flux measurements or too low u*. In 2014 a major instrument failure forced the station to stop measurements in the middle of the season. In 2010 and 2012 there were two more interruptions in the measurements (data gaps of >20 days) although the problems could be solved before the end of the season. Such prolonged gaps led to unreliable gap-filled NEE estimates. REddyProc marginal distribution sampling (MDS) algorithm tended to fill these large gaps with high peaks of respiration at noon times, coercing C uptake underestimation. For this reason, an independent AC NEE dataset (2010-2013) was tested to gap-fill EC data (Figure 3 and Figure S2; supplementary material). The $R^2$ obtained from the EC-AC correlations were always > 0.70 (2010: $R^2= 0.80$, $p < 0.001$; 2011: $R^2= 0.72$, $p < 0.001$; 2012: $R^2= 0.80$, $p < 0.001$; 2013: $R^2= 0.84$, $p < 0.001$). By using AC data, the proportion of missing data was reduced to 28% and we found that the random uncertainty from the combination of AC and MDS algorithm decreased 5% on average. By using the u*filtering and the AC data together with EC, there was an increase of ~6 % in terms of C sink strength. Moreover, the propagated uncertainty in NEE never exceeded ±1.8 g C m$^{-2}$, mainly because the error related to u* filtering was low. Further, we hypothesized that different flux partitioning approaches would lead to different estimates of GPP and $R_{eco}$, however, the results suggest a relatively good agreement (Figure 4). There

was a higher degree of agreement with regard to GPP ($R^2 = 0.83$) compared with $R_{eco}$ ($R^2 = 0.30$). LRC tended to estimate 12 % and 15 % larger GPP and $R_{eco}$, respectively, compared to REddyProc.

## 3.3 Inter-annual and seasonal variation of $CO_2$ ecosystem fluxes

Overall, land-atmosphere $CO_2$ exchange measured for the snow free periods of 2008-2015, omitting 2011, acted as a sink of $CO_2$, taking up -30 g C m$^{-2}$ on average (range -17 to -41 g C m$^{-2}$) (Figure 5; Table 2). The cumulative NEE showed a characteristic pattern during the measurement period (Figure 5), with an initial loss of carbon in early spring right after snowmelt (also observed in Figure 3), followed by an intense C uptake as assimilation exceeded respiratory losses, triggered by increases in temperature, PAR and vegetation growth. This transition point matched the growing season start, when NEE

switched from positive values (a net C source) to negative values (a net C sink). Eventually, the ecosystem turned again into a net C source, defining the growing season end. Even with high inter-annual variability in terms of the end of snowmelt time and growing season start/length (Table 1), the results do not show a marked meteorological effect on the NEE. The ranges in annual GPP (-182 to -316 g C m$^{-2}$) and $R_{eco}$ (144 to 279 g C m$^{-2}$) (Table 2) were >5 fold larger and more variable (CV are 3.6 and 4.1 % respectively) than for NEE (0.7 %). There was a tendency towards larger GPP and $R_{eco}$ during

warmer and wetter years (Figure S3; supplementary material), but there were no warmer and drier years during the study period. The strongest growing season $CO_2$ uptake occurred in 2012 (NEE = -74.2 g C m$^{-2}$; $GS_{length}$ = 78 days), followed by 2010 (NEE = -70.0 g C m$^{-2}$; $GS_{length}$ = 85 days) (Tables 1 and 2). A lengthening of the growing season did not increase the net carbon uptake in this study. In other words, an earlier end of the snowmelt resulting in a longer growing season length did not lead to a stronger carbon sink.

The anomalous year, 2011, constituted a relatively strong source for $CO_2$ (41 g C m$^{-2}$) and was associated with a major pest outbreak, which reduced GPP more strongly than $R_{eco}$. The larvae of the moth *Eurois occulta* data, collected from pitfall traps in the surrounding *Salix* and *Empetrum* dominated plots, showed a strong peak at the beginning of the 2011 growing season (Lund et al., 2017) coinciding with high NEE and very low GPP (Figure 4). In 2011 up to 2078 larvae were observed while other years only 14 (2008), 82 (2009), 186 (2010), 0 (2012) and 8 (2013). It is likely that the reduced primary

production in the wetland area was a partial response to the *Eurois occulta* outbreak.

The daily aggregated NEE-GPP relationships displayed consistent linear correlation (2008-2015: $R^2 = 0.77$, p < 0.001) across the assessed years (Figure 6a). The linear correlations were weaker in 2010 and 2011. A hysteresis was detected in 2010 (i.e. long growing season with higher $R_{eco}$ in autumn compared to spring), while strong C releases was observed in 2011 across June and July. The relation between GPP and $R_{eco}$, which can be understood as the degree of coupling between inputs and

outputs of C, and therefore the degree of C sink strength, showed non-linear patterns (Figure 6b). The curved behaviour is likely because GPP increased more than $R_{eco}$ during early growing season, except for in 2011. Moreover, $R_{eco}$ lagged behind GPP due to (1) the vegetation green-up in the first part of the growing season and (2) the higher respiration rates due to increased biomass in the second part. The years with clearer hysteresis coincide with the years with positive temperature anomalies (i.e. 2010, 2012 and 2013) of the 2008-2015 series. It is worth mentioning the different direction (clockwise vs

counter-clockwise) in the hysteresis observed these years between June, July and August. The data suggest that the clockwise 2012 hysteresis was due to greater gross C cycling (GPP and $R_{eco}$) in June and July favored by warmer conditions; while in 2010 (counter-clockwise hysteresis), the higher gross C fluxes have been measured in August with warmer and wetter conditions (Figure S4; supplementary material).

## 3.4 Environmental forcing

The varied importance of meteorological variables (such as PAR, $T_{air}$, VPD and Precipitation) obtained from Random Forest at different temporal scales (hourly, daily, weekly and monthly) showed differences in behaviour depending on the time aggregation utilized (Figure 7). PAR dominated NEE and GPP while $T_{air}$ correlated the most with $R_{eco}$ in hourly averages,

whereas $T_{air}$ became increasingly important at longer temporal aggregations for all the fluxes (Figure 7). VPD and precipitation were not as important as the other variables while the use of water table depth in the analysis was discarded due to its very low impact on $CO_2$ fluxes. In general, NEE and GPP showed similar distributions of importance, reinforcing the linear relationships found between NEE and GPP (Figure 6). The standard deviation of the variables' importance (across 1000 decision trees) tended to increase at coarser time aggregations.

Changes of environmental forcing (PAR, $T_{air}$ and VPD) across diurnal, seasonal and annual time scales reveal patterns of functional responses to C fluxes. The diurnal cycle analyses on hourly data showed the changes in importance between day- and night-time (Figure 8). NEE and GPP had two predominant variables ($T_{air}$ and PAR) determining the variability at day-time. PAR was important at dawn (06 h. WGST) and dusk (20 h. WGST), while $T_{air}$ was more important at other times. This performance indicates a threshold response to PAR, and a more continuous response to temperature. On the other hand, $R_{eco}$ was mainly driven by $T_{air}$ at both night-time and day-time. VPD and PAR had a negligible impact on $R_{eco}$. The seasonal pattern importance showed PAR dominating NEE and GPP from early June to early October (Figure 8), while $T_{air}$ and VPD became more important before and after the snow-free conditions. In terms of $CO_2$ emission ($R_{eco}$) the pattern is less clear and noisier, although $T_{air}$ appeared to be the most important variable. Finally, the annual pattern exposes a performance in line with previous results, i.e. PAR dominated NEE and GPP while $R_{eco}$ was more sensitive to variations of $T_{air}$. Interestingly, the Random Forest analysis revealed a decrease of PAR's importance in 2011, same year exposing the sharp decrease of C sink strength.

## 4 Discussion

### 4.1 Data processing and quality

The NEE gap-filling and subsequent partitioning into GPP and $R_{eco}$ are needed to understand the $CO_2$ flux responses to the environmental forcing. However, these procedures expose unavoidable uncertainties in the seasonal C budget calculation (Table 2) and partial inconsistencies between approaches (Figure 4). In this study, we used a marginal distribution sampling (MDS) gap-filling technique, an enhancement to the standard look up table (LUT). Both methods have shown a good overall performance compared to other procedures such as non-linear techniques (NLRs) or semi-parametric models (SPM), but slightly inferior to artificial neural network (ANN) (Moffat et al., 2007). However, the MDS gap-filling alone introduced NEE estimates out of range across the two extensive gaps in 2010 and 2012 (Figure S2; supplementary material). Quantifying the uncertainty introduced by measurement gaps is complex (Falge et al., 2001; Moffat et al., 2007; Papale et al., 2006). One possibility would be a sensitivity analysis of time series with artificially introduced gaps (Dragomir et al., 2012; Pirk et al., 2017). But the choice of gap length and position is difficult, and would render uncertainty to the uncertainty assessment itself. Instead, we used the EC prediction based on independent auto-chamber (AC) measurements between 2010 and 2013. The agreement between EC and AC were always $R^2 > 0.72$ and $p < 0.001$, and the 95% confidence interval of the predictions were reported together with the resulting uncertainties (Table 2). Although the AC data itself incorporated a new source of uncertainty to the calculations, we consider this method to be less weak than an unreliable gap-filling estimate. We used the AC as platform to decrease the gap length and the total random uncertainty (Aurela et al., 2002) before the MDS algorithm was applied. AC was used together with MDS, and never was used as an independent gap-filling procedure.

The NEE partitioning obtained from REddyProc and LRC suggests a relatively good agreement in model performance. The one-to-one comparison between different approaches found a better agreement with regard to GPP compared to $R_{eco}$. In this analysis, REddyProc produced smoother $R_{eco}$ estimates compared to the noisier GPP estimates, whereas LRC performed the other way around. This is mainly because measurement noise goes into GPP for REddyProc method, and into $R_{eco}$ for LRC method. REddyProc retrieves positive GPP values whereas LRC method results in negative $R_{eco}$ values. Both scenarios are not fully convincing, although it is not straightforward how they should be treated. By removing all positive GPP / negative

$R_{eco}$ values would risk removing only one side of the extremes. Besides night-time based (REddyProc) and day-time based (LRC) partitioning approaches, several implementations have been proposed to improve the algorithms performance. Lasslop et al. (2010) has modified the hyperbolic LRC to account for the temperature sensitivity of respiration and the VPD limitation of photosynthesis. Further, Runkle et al. (2013) proposed a time-sensitive multi-bulk flux-partitioning model, where the NEE time series was analyzed in one-week increments as the combination of a temperature-dependent $R_{eco}$ flux and a PAR-dependent flux (GPP). However, it remains uncertain under which circumstances each partitioning approach is more appropriate, especially in the boundaries between low- and high-Arctic due to the lack of dark night during polar days (when light is not a limiting factor for plant growth). Since there are few methods with an unclear precision, an evaluation study on the effect of using different partitioning approaches along latitudinal gradients would be very beneficial to assess the suitability for each method.

### 4.2 Inter-annual and seasonal variation of $CO_2$ ecosystem fluxes

The balance between the two major gross fluxes in terrestrial ecosystems, photosynthetic inputs (GPP) and respiration outputs ($R_{eco}$), displayed larger temporal variability than did NEE. These results suggest that both GPP and $R_{eco}$ were strongly coupled and sensitive to meteorological conditions such as insolation and temperature (Figure 7 and 8). Interestingly, the tendency to warmer and wetter conditions led to greater rates of C cycling associated with larger GPP and $R_{eco}$ (Figure S3; supplementary material). This result does not entirely coincide with Peichl et al. (2014), even though they performed a similar analysis for a Swedish boreal fen. This finding points towards the complexity in the response of wetland ecosystems towards changing environmental conditions. The response is dependent on many things, such as hydrological settings, and these differ between sites. In this study, larger rates of C uptake (GPP) were linked to larger rates of C release ($R_{eco}$), with the exception of the anomalous year 2011. The relative insensitivity of NEE to meteorological conditions during the snow-free period could be the result of the correlated response of ranked cumulative GPP and $R_{eco}$ (Figure 5) (Richardson et al., 2007; Wohlfahrt et al., 2008). This site likely receives more precipitation relative to many other tundra ecosystems, and has no permafrost, thus the NEE response to climate could be less variable. However, as Kobbefjord is located in a coastal area, it is not surprising to receive high precipitation, and other ecosystems such as coastal blanket bogs often receive even more precipitation without a clear impact of drought effect on the NEE sensitivity (Lund et al., 2015). Furthermore, permafrost adds another layer of complexity to the C dynamics (Christensen et al., 2004; Koven et al., 2011; Schuur et al., 2015). Although some studies showed similarities of $CO_2$ fluxes in various northern wetland ecosystems with and without permafrost (Lund et al., 2015), permafrost has strong influence on the hydrology of peatlands (Åkerman and Johansson, 2008), and therefore their topography and distribution of vegetation (Johansson et al., 2013). Especially in the context of climate warming permafrost thaw can cause large changes to the ecosystems. Further, this study agrees with Parmentier et al. (2011) and Lund et al. (2012), who suggested that a longer growing season does not necessarily increase the net carbon uptake. Here a more negative NEE indicated a stronger C sink (i.e.) in 2012 compared to 2010. Parmentier et al. (2011) hypothesized that this behavior is due to site-specific differences, such as meteorology and soil structure, and that changes in the carbon cycle with longer growing seasons will not be uniform around the Arctic. Thus, the effects of climate change on the tundra C balance of are not straightforward to infer.

NEE measured in Kobbefjord from 2008 to 2015 indicates a consistent sink of $CO_2$ (within a range of -17 to -41 g C m$^{-2}$) with exception of the year 2011 (+41 g C m$^{-2}$) (Table 2). The year 2011, associated with a major pest outbreak, reduced GPP more strongly than $R_{eco}$ (Figure 5) and Kobbefjord turned into a strong C source within an episodic single growing season. The return to a substantial cumulative $CO_2$ sink rates following the extreme year of 2011 shows the ability of the ecosystem to recover from the disturbance (Lund et al., 2017). Indeed, the ecosystem not only shifted back from being a C source to a C sink, but it also changed rapidly from one year to the next. Thus we found evidence in Kobbefjord of ecosystem resilience to the meteorological variability, similar to other cases described in other northern sites (Peichl et al., 2014; Zona et al., 2014).

Only a few reference sites have reported similar decreases in net C uptake, but in no case as large as the one observed here. Zona et al. (2014) described an effect of delayed responses to an unusual warm summer in Alaska. Their results suggested that vascular plants, which have enhanced their physiological activity during the warmer summer, might have difficulties readapting to cooler, but not atypical, conditions, which have provoked a significant decrease of GPP and $R_{eco}$ the following year. In their study, the ecosystem returned to be a fairly strong C sink after two years, suggesting strong ecosystem resilience. Moreover, Hanis et al. 2015 have reported comparable C sink - C source variations in a Canadian fen within the growing season due to changes in the water table depth. Drier and warmer than normal conditions have triggered an increase in C source strength. Finally, during an extensive outbreak of autumn and winter moths in a subarctic birch forest in Sweden, Heliasz et al. (2011) observed a similar decrease in net sink of C (most likely due to weaker GPP) across the growing season. However, the C source strength (NEE = 40.7 g C m$^{-2}$) found in 2011 at Kobbefjord was higher compared to these other cases. To our knowledge, such abrupt disturbance concerning C sink strength in Arctic tundra has not be previously reported excluding severely burned landscapes (Rocha and Shaver, 2011).

A combination of different factors could have led to the sharp change in C balance observed between 2010-2011, both physical and biological. The year 2010 had the warmest mean annual temperature (3.4 °C compared to the -0.4 °C mean annual temperature for 2008-2015) and the warmest mean wintertime temperature (-2.7 °C compared to the -6.79 °C mean for 2008-2015) (Figure 2a). These climatic conditions generated the thinnest (maximum daily snow depth of 0.3 m compared to averaged 0.9 m) (Table 1) and shortest-lasting snowpack. Consequently, 2010 had the longest growing season (85 days) and very high growing season C uptake (-70 g C/ m$^{-2}$). Increases in temperature can lead to high respiration rates during early winter (Commane et al., 2017; Zona et al., 2016), but also during the following summer (Helfter et al., 2015; Lund et al., 2012), which is related to soil temperature and snow dynamics. Further, in Kobbefjord the year 2011 had one of the lowest mean annual temperatures and mean wintertime temperatures (-1.7 and -6.1°C respectively), which created the thickest (maximum daily snow depth of 1.4 m) and the longest-lasting snowpack, leading to the shortest growing season for the study period (only 47 days). According to Lund et al. (2012), below thick snowpack soils will be insulated from reaching low temperature, acting as lid and preventing $R_{eco}$ from being released to the atmosphere until the snowmelt period. Finally, larvae of the noctuid moth *Eurois occulta* outbreak occurred in 2011, overlapping the abrupt decrease of C sink strength observed. Although we cannot provide a quantification of change attributed to meteorological variations and biological disturbances, there is evidence showing that the moth outbreak could partially have decreased the C sink strength in Kobbefjord. In an undisturbed scenario, the meteorological conditions in 2015, colder and dryer than the mean 2008-2015 period (Figure 2), but similar to 2011, would have stimulated similar behaviours in terms of C fluxes. However, the cumulative fluxes in 2015 (Figure 5) followed analogous patterns compared to the rest of the years. This evidence agrees with literature (Callaghan et al., 2012b; Lund et al., 2017) on the fact that tundra systems can fluctuate in sink strength influenced by factors such as episodic disturbances or species shifts, events very difficult to predict.

**4.3 Environmental forcing**

Our data indicates that the importance of the main environmental controls (radiation and temperature) for C fluxes did vary across diurnal, seasonal and annual cycles, but also between time aggregations. The hourly variability of NEE and GPP (Figures 7 and 8) was mostly dependent on PAR because of the threshold nature on radiation control on GPP. Overall, the results indicate that environmental factors that can change rapidly such as PAR will have a high influence on short time scales (Stoy et al., 2014). The increased importance of PAR at 08 h and 20 h WGST coincides with the sharp gradient in light at dawn and dusk (Figure 8). The control of PAR on GPP is not a new finding itself, but the Random Forest approach helps to quantify its importance. There is no GPP at night, and therefore there will be a strong increase/decrease in GPP at dawn/dusk. The seasonal pattern also showed that radiation is the single main driver for NEE and GPP between early June and early October, supported by the longer day-time. Further, PAR appeared to be a limiting factor for annual NEE in 2011,

increasing further the complexity around this anomalous year. These results agree with literature (Groendahl et al., 2007; Stoy et al., 2014) suggesting that the uptake of $CO_2$ is partially controlled by radiation for the photosynthetic physiology at the leaf scale. Arctic plants are usually well adapted to environments with low light levels, reporting near-maximum rates ranging from 10°C to 25°C (Oechel and Billings, 1992; Shaver and Kummerow, 1992).

Photosynthesis is restricted by low temperature, so enzymatically driven processes such as carbon fixation are more sensitive to low temperature than the light-driven biophysical reactions (Chapin et al., 2011). In this paper the daily, weekly, and monthly aggregated variability of C fluxes was primarily linked to $T_{air}$. Moreover, the Random Forest analyses revealed a strong diurnal pattern with a marked contribution of $T_{air}$ to variations in NEE and GPP (both at night-time and between 08-18 h WGST). These results agree with Lindroth et al (2007), who recognized $T_{air}$ as key driver for NEE seasonal trends in northern peatlands. However, in this analysis both NEE and GPP had similar responses to common environmental forcing, contrary to the results in Reichstein et al. (2007). In order to circumvent the potential circularity conflicts based on the use of partitioning products, we filtered day-time NEE (true GPP) and night-time NEE (true $R_{eco}$), obtaining very similar results (Table S2; supplementary material). Further, our data also suggest that $R_{eco}$ is often dominated by air temperature. The patterns observed here are in agreement with findings on plant respiration dynamics (Heskel et al., 2016; Lloyd and Taylor, 1994; Tjoelker et al., 2001).

In this study, environmental drivers related to water availability such as VPD and precipitation were not found to be as influential as other assessed variables. We did not find significant relationships between $CO_2$ fluxes and the water table depth. Thus, there was no apparent water limitation on carbon dynamics during the eight years period. However, the complex interactions based on changes in temperature and soil moisture particularly over full annual cycles and for sites with permafrost, should be further explored. Our results contrast with Strachan et al. (2015) who described water table depth as an important driver regulating the $CO_2$ balance and others who found that $CO_2$ emissions increase during dry years due to increased decomposition rates and a reduction in GPP (Aurela et al., 2007; Lund et al., 2007; Oechel et al., 1993; Peichl et al., 2014); whereas other sites act as sinks during relatively wet years (Lafleur et al., 1997). The fen in Kobbefjord is probably quite resistant to droughts since it is fed with water from the surroundings.

**5 Conclusions**

We have analyzed eight snow-free periods in eight consecutive years in a West Greenland tundra (64° N) focusing on the net ecosystem exchange (NEE) of $CO_2$ and its photosynthetic inputs (GPP) and respiration outputs ($R_{eco}$). Here, the NEE gap-filling exposed inherent uncertainties in the seasonal C budget calculation, but there were also inconsistencies between the flux partitioning approaches used. We find that Kobbefjord acted as a consistent sink of $CO_2$, during the years 2008-2015, except 2011 that was associated with a major pest outbreak. The results do not show a marked meteorological effect on the net C uptake. However, the relative insensitivity of NEE during the snow-free period was driven by the correlated, balancing responses of GPP and $R_{eco}$, both more variable than NEE and sensitive to temperature and insolation. In this paper we show a tendency towards larger GPP and $R_{eco}$ during wetter and warmer years. The anomalous year 2011, affected by a biological disturbance, constituted a relatively strong source for $CO_2$ and reduced GPP more strongly than $R_{eco}$. A novel analysis assessing the changes of environmental forcing across diurnal, seasonal and annual time scales unmasked patterns of functional responses to C fluxes.

Despite the fact that we analysed an eight-year dataset, the results do not provide a complete picture due to the lack of year round data (Grøndahl et al., 2008). The snow season should be taken into account for a comprehensive understanding of complete C budget (Aurela et al., 2002; Commane et al., 2017; Zona et al., 2016) and the delayed effect of wintertime-based variables such as snow depth or snow cover on the C fluxes. Because some studies have suggested that GPP and $R_{eco}$ have increased with observed changes in climate and NEE trends remain unclear (Lund et al., 2012), it is challenging to produce

strong evidence while the data remains scarce and fragmented. Hence, there is a need for increased efforts in monitoring of Arctic ecosystem changes over the full annual cycle (Euskirchen et al., 2012; Grøndahl et al., 2008). Future work is also required with C flux modelling in order to explore into process-based insights of C exchange balance in the Arctic tundra, and the interactions of photosynthesis and $R_{eco}$ with changes in C stocks.

**Author contribution**

E. López-Blanco, M. Lund, M. Williams, T.R. Christensen and M.P. Tamstorf designed the experiment. Data preparation and analysis was primarily performed by E. López-Blanco with contribution from M. Lund (eddy covariance data processing, data quality control and LRC partitioning), M. Williams and T.R. Christensen (experimental set up), B.U. Hansen (data gathering from Nuuk Ecological Research Operations, GeoBasis), A.Westergaard-Nielsen (daily estimate of the timing of snowmelt) and J.-F. Exbrayat (Random Forest approach). E. López-Blanco prepared the manuscript with active contributions from all co-authors.

**Acknowledgements**

This work was supported in part by a scholarship from the Aarhus-Edinburgh Excellence in European Doctoral Education Project and by the eSTICC (eScience tools for investigating Climate Change in Northern High Latitudes) project, part of the Nordic Center of Excellence. The authors wish to thank the Nuuk Ecological Research Operations (nuuk-basic.dk) as well as GeoBasis program, which is in charge of the eddy covariance and the auto-chamber systems. Both projects are being run under the Greenland Ecosystem Monitoring (GEM) program funded by the Danish Environmental Protection Agency and the Danish Energy Agency.

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

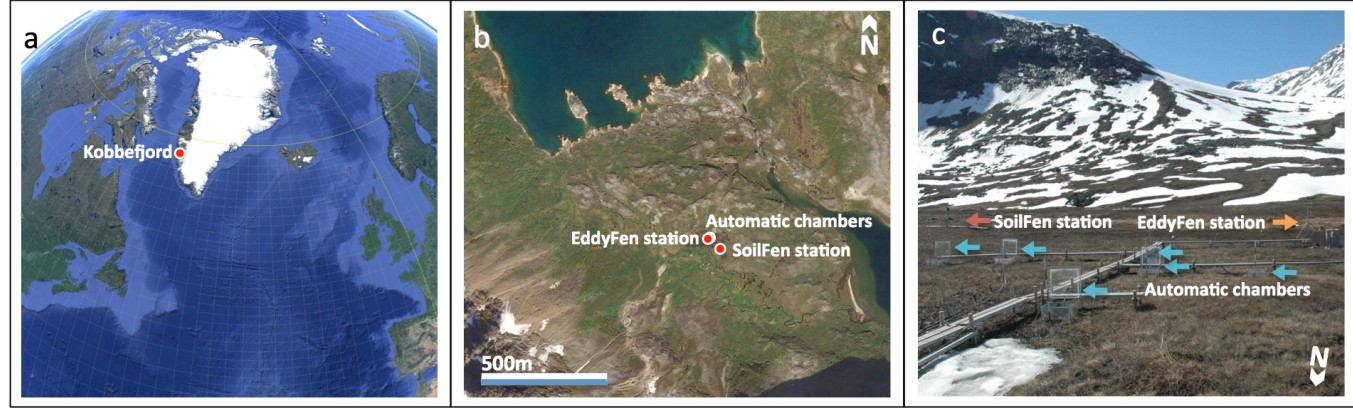

**Figure 1:** (a) Location of Kobbefjord in Greenland, 64° 07' N; 51° 21' W (Source: Google Earth Pro). (b) Location of EddyFen station, automatic chambers and SoilFen station in Kobbefjord (Source: Google Earth Pro, 16-07-2013). (c) Eddy covariance (orange arrow) from EddyFen station, six automatic chambers (light blue arrows) and SoilFen station (pale red arrow)(photo by Efrén López Blanco, 27-06-2015).

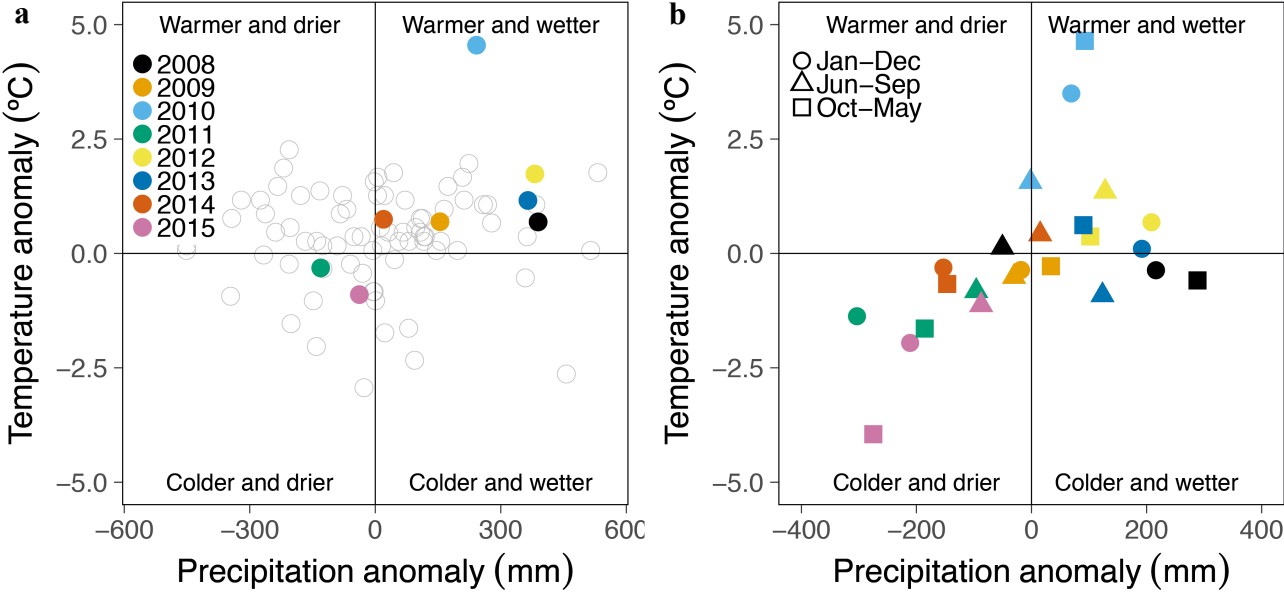

**Figure 2:** (a) Annual Temperature (°C) and precipitation (mm) anomalies of the analyzed years (2008-2015) compared to the 1866-2007 time series shown as empty circles (Cappelen, 2016), and (b) within the 2008-2015 period including annual (January to December), warm season (July to September) and cold season (October to May) averages.

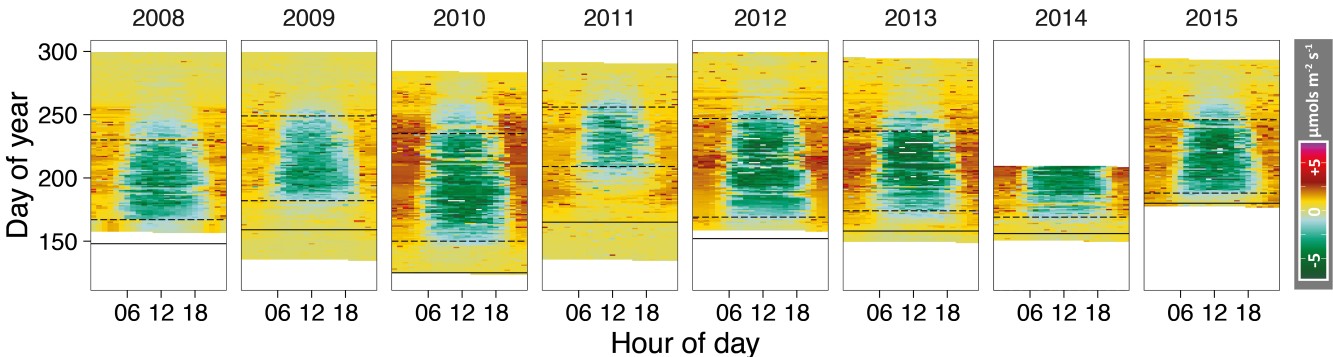

**Figure 3.** Time series of gap-filled NEE (2008-2015) based on auto-chamber data (2010-2013) and the MDS algorithm (from REddyProc). Green represents C uptake while the orange-dark red denotes C release. The solid lines represent the end of the snowmelt period while the area within the dashed lines represent the period between the start and the end of the growing season.

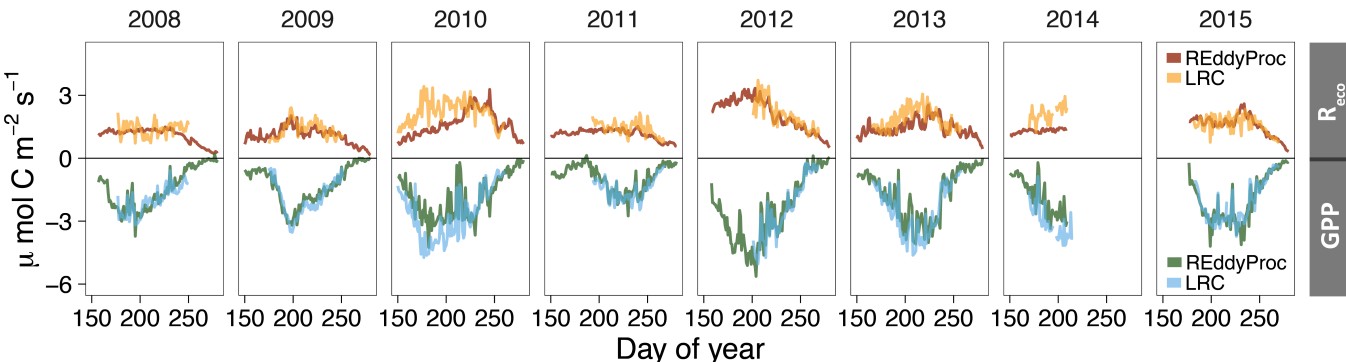

Figure 4. Time series of daily mean GPP (negative fluxes) and R$_{eco}$ (positive fluxes) from 2008 to 2015 calculated by REddyProc (dark green and dark red) and LRC (orange and light blue).

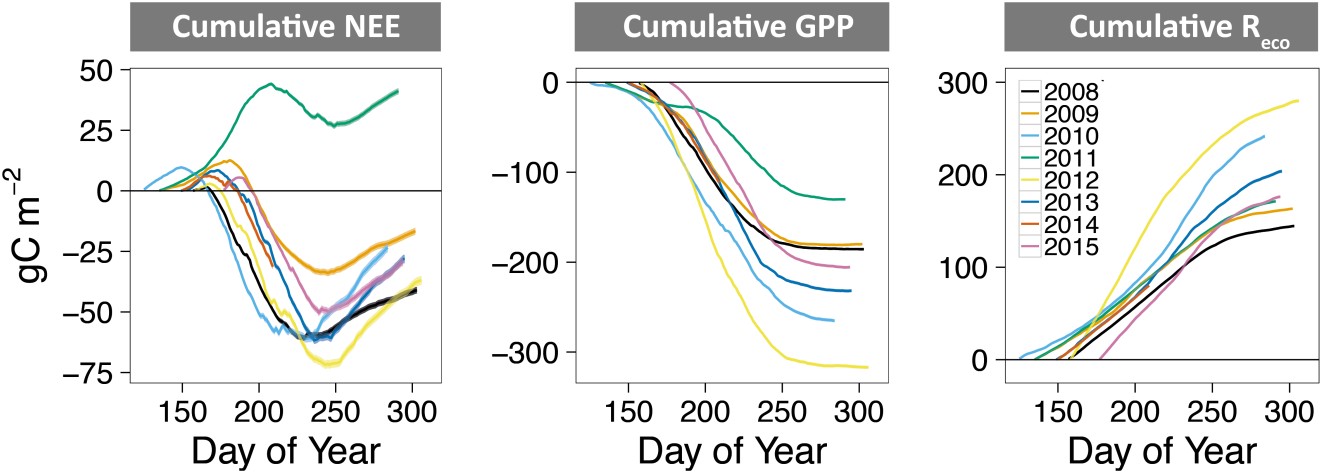

Figure 5. Cumulative NEE, GPP, and R$_{eco}$ from 2008 through 2015 including the u* filtering and random errors.

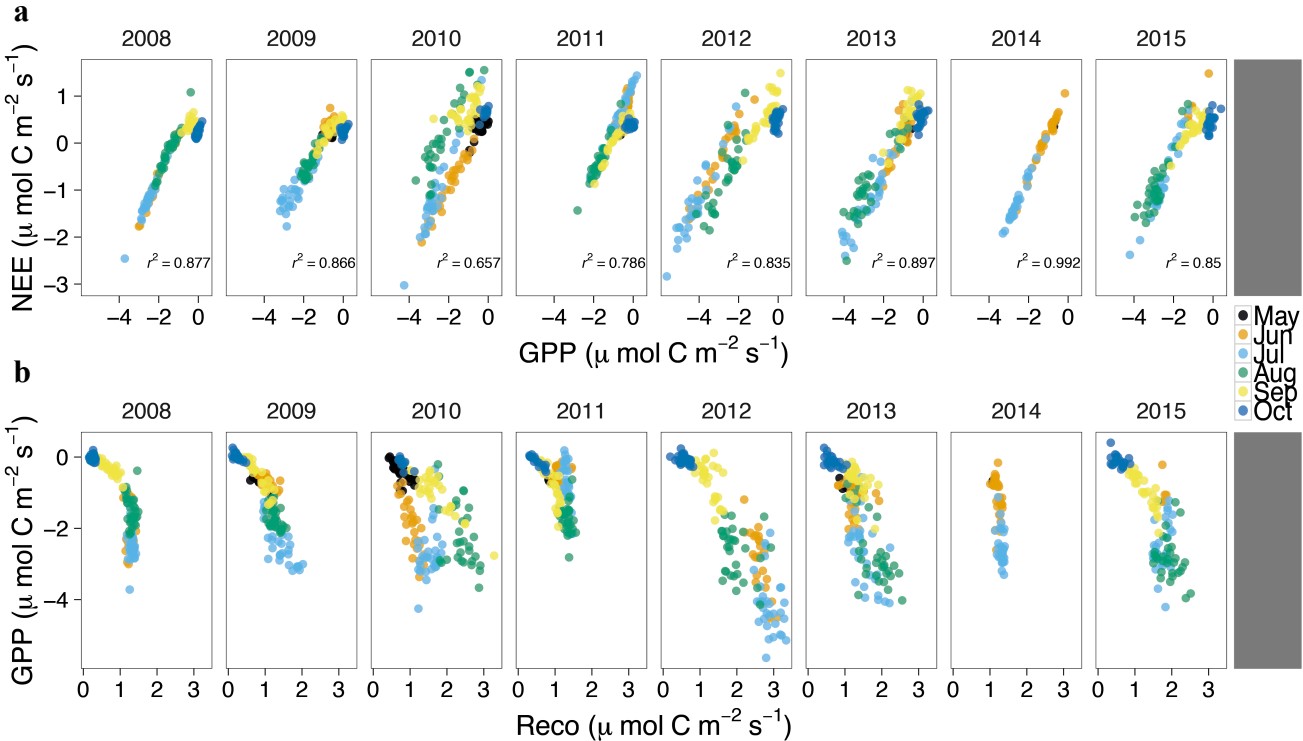

Figure 6. Inter-annual variability between (a) NEE-GPP and (b) GPP-R$_{eco}$ relationships. The data was daily aggregated and colored per month

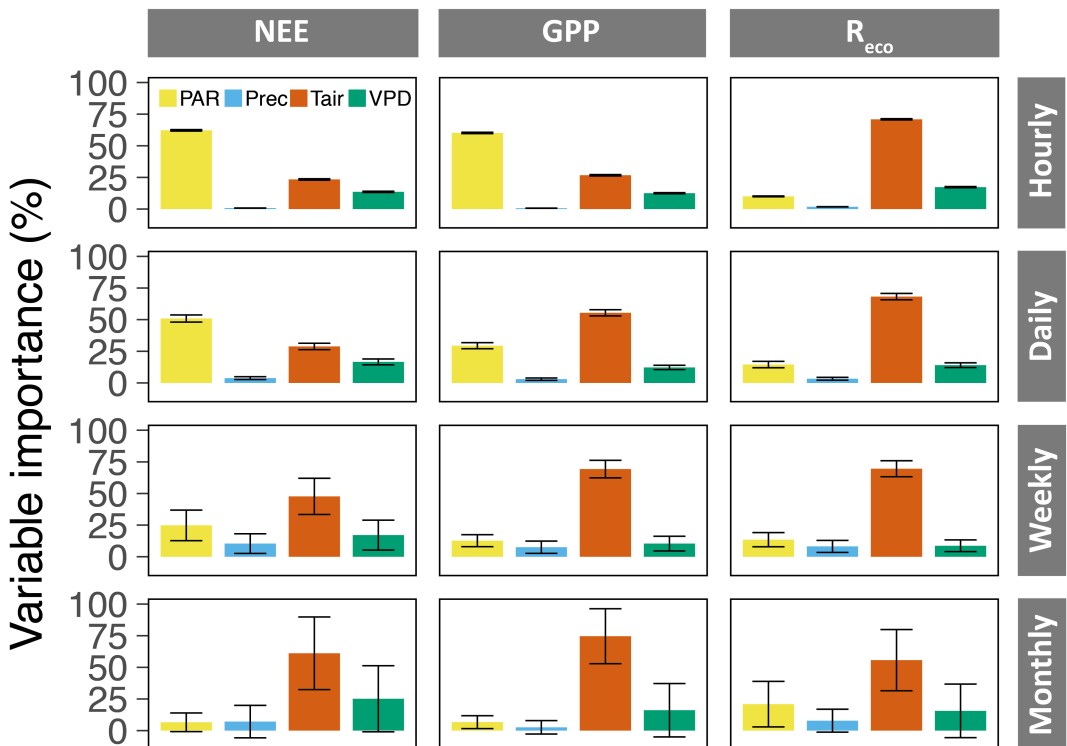

**Figure 7. Importance of environmental variables PAR (yellow), $T_{air}$ (orange), Prec (pink) and VPD (green) to explain variability in NEE, GPP and $R_{eco}$ (partitioned by REddyproc) at different temporal aggregations (hourly, daily, weekly and monthly) when all the years were lumped together. Thick bars and error bars represent the mean ± standard deviation of the importance across 1000 decision trees.**

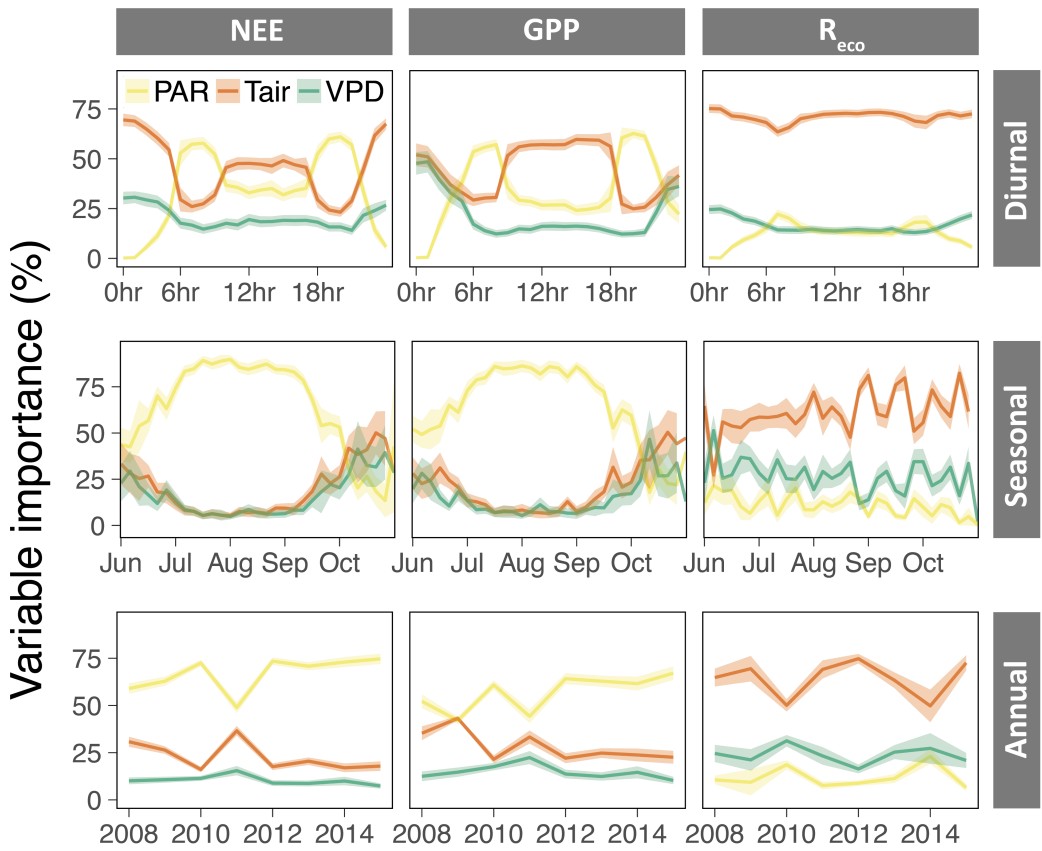

**Figure 8. Diurnal, seasonal and annual importance of environmental variables PAR (yellow), $T_{air}$ (orange), and VPD (green) to explain variability in NEE, GPP and $R_{eco}$. Thick lines and shading represent the mean ± standard deviation of the importance across 1000 decision trees.**

**Table 1. Summary of the phenology-related variables for the period 2008-2015.**

| | 2008 | 2009 | 2010 | 2011 | 2012 | 2013 | 2014 | 2015 |
|---|---|---|---|---|---|---|---|---|
| Maximum snow depth (m) | 0.6 | 1.0 | 0.3 | 1.4 | 1.0 | 0.6 | 1.1 | 1.2 |
| End of snowmelt period (DOY) | 148 | 159 | 125 | 165 | 152 | 158 | 156 | 176 |
| Beginning of growing season (DOY) | 167 | 182 | 150 | 209 | 169 | 174 | 169 | 188 |
| End of growing season (DOY) | 230 | 249 | 235 | 256 | 247 | 237 | - | 246 |
| Length of growing season (DOY) | 63 | 67 | 85 | 47 | 78 | 63 | - | 58 |

755 **Table 2. Summary of the measuring periods and the growing season CO2 fluxes for the period 2008-2015.**

| | 2008 | 2009 | 2010 | 2011 | 2012 | 2013 | 2014 | 2015 |
|---|---|---|---|---|---|---|---|---|
| First measurement (DOY) | 157 | 135 | 124 | 135 | 158 | 149 | 150 | 177 |
| Last measurement (DOY) | 303 | 304 | 282 | 287 | 305 | 295 | 209* | 294 |
| Missing data (%) | 57.6 | 42.3 | 28.6 | 35.4 | 32.3 | 29.8 | 44.9* | 40.0 |
| NEE in measuring period (g C m$^{-2}$) | -41.3 | -16.9 | -24.4 | 40.7 | -37.0 | -28.1 | -28.7* | -31.5 |
| | ±1.4 | ±1.4 | ±1.9 | ±1.3 | ±1.8 | ±1.7 | ±1.1 | ±1.6 |
| NEE in growing season (g C m$^{-2}$) | -62.3 | -45.9 | -70.0 | -16.2 | -74.2 | -69.7 | -35.3* | -55.8 |
| Maximum daily uptake (DOY) | 195 | 205 | 182 | 230 | 204 | 220 | 192* | 199 |
| Maximum uptake ($\mu$mols m$^{-2}$ s$^{-1}$) | -2.4 | -1.7 | -3.0 | -1.4 | -2.8 | -2.5 | -1.9* | -2.3 |
| Estimated GPP (g C m$^{-2}$) | -185.5 | -181.8 | -266.1 | -130.6 | -316.2 | -230.7 | -106.8* | -206.1 |
| | ±1.4 | ±1.4 | ±1.9 | ±1.3 | ±1.9 | ±1.7 | ±1.1 | ±1.6 |
| Estimated R$_{eco}$ (g C m$^{-2}$) | 144.2 | 164.9 | 241.6 | 171.3 | 279.2 | 202.6 | 78.1* | 174.6 |
| | ±1.3 | ±1.3 | ±1.8 | ±1.2 | ±1.8 | ±1.7 | ±1.1 | ±1.5 |

where applicable: ± sum of the auto-chamber, random and u* filtering uncertainties, * incomplete growing season dataset.