# Peer review of "Exchange of CO2 in Arctic tundra: impacts of meteorological variations and biological disturbance"

_Biogeosciences, 2016_

## Referee Comment (RC1) · Anonymous Referee #1 · 17 May 2017

Lopez-Blanco and colleagues present a study of ecosystem CO2 dynamics across eight snow-free seasons for a wet fen tundra ecosystem in west Greenland. The authors compare ecosystem respiration (Reco) and gross primary production (GPP) with key climatic drivers to characterizes how ecosystem CO2 dynamics will change with climate. Comparisons are made at hourly, daily, and seasonal timescales to understand how drivers of ecosystem CO2 dynamics change across temporal scales. Additionally, the authors compare several eddy covariance partitioning methods in order to assess uncertainty associated with interpretation of EC derived estimates of Reco and GPP. The main finding is that large interannual variations in Reco and GPP with climate are compensatory, and so net ecosystem exchange (NEE) of CO2 remains quite stable

across climatically diverse snow-free seasons. This is a valuable analysis of a fairly long EC data set, particularly for a tundra ecosystem. Overall I find the methodology to be quite sound and recommend several relatively minor but important revisions before the manuscript is considered further for publication. The following paragraphs describe more major issues, and are then followed by specific comments.

The introduction should be improved in several ways. First, the paragraph on flux partitioning seems out of place. The first and third paragraphs highlight research surrounding tundra/Arctic C cycling, and are bisected by the paragraph on partitioning. It would make more sense to first discuss carbon cycle dynamics and then highlight challenges associated with EC partitioning; so switch paragraphs two and three.

In the results it seems that sections 3.3 should come before section 3.2; first describe the partitioning comparisons and then get into the results. Related, I don't see where you mention which partitioning/gapfilling methods you report. It would make sense to first present the flux processing results, and then state which date you'll present moving forward. Also, it is general good to have the figures ordered as they appear in the text. Currently order is Fig 5 -> Fig 4 -> Fig 3.

The last major area for revision is related to the broader implications of your results – specifically, how transferable are they? There is some of this in section 4.3, but it could be expanded there, and perhaps in section 4.1. Specifically, it occurs to me that this research site receives a relatively high amount of precipitation relative to many other tundra ecosystems, and has no permafrost. As such, the NEE responses to climate at other tundra sites may likely be more variable. It would be worth discussing this a bit further. Secondly, it is difficult to talk about ecosystem $CO_2$ source/sink dynamics without some discussion of non-growing season processes. Papers by Zona et al and Commaine etal (very recently) indicate the importance of non-growing season C dynamics. Also, given the fact that you are using net sink timing to define the growing season, I wonder what effect previous growing season or previous winter conditions might have on your results? For example does a wet summer followed by a warm winter
lead to high Reco the following year? There are very likely some interesting time-lag effects influencing the patterns you observe. Again, you allude to these processes, for example, by mentioning previous winter temperatures, but I think a more targeted and thoughtful discussion on temporal lags/dynamics would be useful. Actually, it would be helpful to report non-growing season climate data, and perhaps even analysis of these sorts of time lags. I do not think the latter is absolutely necessary, because this paper already contains a lot of information, but it could be informative either here or in another paper.

(I will also note here that it seems odd to place the section on EC processing between to two sections discussing CO2 dynamics).

Minor edits:

Lines 40-44: You should explicitly state that you are referring to soil C stocks – this doesn't come until the very end.

Line 76: Why do you mention C a need for sites with C stocks if you don't present them in the paper?

Line 102: This line is a bit too informal; it's not Skip's map, it was a large collaborative effort. It would be more appropriate to report the class and the name of the map and the paper describing the map. Walker, D. et al. (2005), The Circumpolar Arctic vegetation map, Journal of Vegetation Science, 16(267-282).

Lines 103-104: I don't understand this, what does it mean that the site 'went out of the Arctic zone'?

Line 142: What is Papale et al In Prep? Perhaps indicate that this is via personal communication as well, if that is the case.

Line 264: This is a very simplistic and incomplete view of the residence time of fixed C. I'm not sure you can say anything meaningful about C residence time with discussing fluxes between pools and storage, which aren't really addressed in this manuscript.

Line 279: This could be worded clearer; at first I thought you were saying the PAR values peak at 6am, which was confusing. Perhaps explicitly state that the predictive importance of PAR peaks at this time.

Line 287: The model 'catching' something is perhaps a bit too colloquial. Better to state that it revealed or indicated a decline in the importance of PAR in 2011.

Line 295: You can only say that NEE is insensitive to climate during the snow-free season.

Line 300: 'NEE exchange' is redundant, just use NEE (here and elsewhere).

Line 330: Lots of typos here.

Figures 4 & 7: It would be good to include a legend indicating what the colors represent, in addition to the text description.

---

## Referee Comment (RC2) · Anonymous Referee #2 · 29 Jun 2017

The article "Exchange of CO2 in Arctic tundra: impacts of meteorological variations and biological disturbance" by Lopez-Blanco and co-authors presents eight years of eddy covariance measurements from a tundra site in Greenland. The data set is rich and the authors apply current and appropriate methods in data analysis to derive gap-filled net carbon fluxes, as well as to partition these fluxes into the photosynthetic and respiration components. The authors attempt to analyze gap-filling procedures and use auto chamber data towards these efforts. The undertaken analyses reveal valuable insight into the behavior of tundra carbon cycling in response to environmental variability from hourly to inter annual scales. Novel methods are applied to analyze the role of environmental drivers of C cycling as well as biological factors such as a pest outbreak. In

general, the manuscript is a solid and valuable contribution. Greater attention to grammar, structure, and clarity will greatly improve the article. In some cases, additional justification for statements or references to literature are needed. The comments that follow provide suggestions for addressing these concerns before publication.

General comments:

There is too much repetition in portions of the manuscript (specific comments identify some of these sections), and efforts to reduce repetition will increase the readability of the paper.

More attention is needed to grammar throughout the manuscript. Importantly please play close attention to the correct use of singular or plural nouns. Here are some examples where they should be switched (but please address on a case by case basis):

Singular case instead: temperatures -> temperature, exchanges -> exchange, budgets -> budget, precipitations -> precipitation, references -> reference, evidences -> evidence

Data: plural -> data are rare

Capitalize Earth and Arctic when proper nouns

With respect to figure 6, what causes the different direction (clockwise vs counterclockwise) in the hysteresis observed in 2010 vs 2012 vs 2013? It would be interesting to know the whether the causes for early versus late season decoupling of GPP and Reco are the same or different.

Specific comments:

Abstract: I find the use of meteorology and climate to be a bit conflicting here. Please ensure whether you mean meteorology or climate with reference your conclusions in this study.

P2L69: The terminology "C balance state" does not carry an immediate clear meaning.

Does this refer to the annual balance of net carbon exchange? Clarify what C state refers to and how relates to fluxes versus carbon stocks and over which time frames. What is your definition of C uptake and C storage, and over what time frame?

P2L52: Eddy covariance data can include other types of gases, so good to specify: Eddy covariance measurements of $CO_2$

P3L82: Resiliency in which sense? Should clarify right away.

P3: Sections of the end of the introduction are too detailed to be placed in the introduction and should be moved to the materials and methods section. Please separate material between L82-91 into intro vs methods as appropriate

P3L116: clarify what 5+5 min means

P3L120: spell out km if used in this sentence

P5L184: Please clarify what is meant by "sums the variable's importance up to 1". This sentence could be clearer

P5L198: Check grammar: "also exposed a larger variability"

P6L205: what is a non-lap year?

P6L216: measurement period

P6L223 & Fig S4: The largest GPP and Reco were found in wetter and warmer years, but what is the statistical measure to support a "tendency towards larger GPP and Reco during wetter and warmer years"? For example, for Reco, half of warmer/wetter are larger and half are smaller than colder/drier.

P6L228: perhaps be more specific about what the response to the outbreak was in terms of fluxes (not really a response of measurements, but of actual fluxes). Just GPP?

P7L281: I wouldn't use "momentarily" to describe hourly data

P7L285: What is meant by "although Tair appeared to be the less limiting factor". It seems that Tair is the most important variable for Reco, but I'm not sure how it would be limiting or not

P8L1286: Check grammar in the last sentence. I wouldn't use "catch". Please elaborate on what the connection here is. Why would a decrease in PAR's importance are sense here?

P8L293: What tendency is that? Also, don't use 'mirror effect'. Use clearer language.

P8L298: I'm not sure this sentence is a natural conclusion from your results: "Thus, the effects on C balance of warming from climate change are not straightforward to infer." Would these processes not be predicted by models? If so, then it could be misleading to state that it is difficult to infer. Provide some context from current literature here if in fact current understanding would have missed this.

P8L303: a bit redundant with 'growing season' twice

P8L314: outbreak of what?

P8L317, L330: check grammar

P8L322: shortest-lasting, longest-lasting

P9L337: This first two sentences are very unclear as written

P9 section 4.2: I don't find this analysis of gap filling to be very informative because estimates regarding which method is best are not testable. Why not test the performance of the gap-filling on years where you have good data coverage by creating artificial gaps and testing model performance against real data? I would find that exercise to be much more compelling and would help you determine which method to apply in years where data is really missing.

P9L365: How was the filtering done? This is not clear.

P10: I would avoid using 'interesting' so much as a way to describe your observations. It would be more informative to put in context with extant literature. You should not just repeat results here that are listed elsewhere, but put into context. For example, this is done in the latter half of the L380-387 paragraph, but not the first part. The first half of the conclusion is a bit repetitive as well - should not be a repetition of abstract, should be more general.

Table S1: Avoid using N°

Where is Figure S1?

---

## Author Comment (AC1) · 21 Jul 2017

**Referee #1**

*Lopez-Blanco and colleagues present a study of ecosystem CO2 dynamics across eight snow-free seasons for a wet fen tundra ecosystem in west Greenland. The authors compare ecosystem respiration (Reco) and gross primary production (GPP) with key climatic drivers to characterizes how ecosystem CO2 dynamics will change with climate. Comparisons are made at hourly, daily, and seasonal timescales to understand how drivers of ecosystem CO2 dynamics change across temporal scales. Additionally, the authors compare several eddy covariance partitioning methods in order to assess uncertainty associated with interpretation of EC derived estimates of Reco and GPP. The main finding is that large interannual variations in Reco and GPP with climate are compensatory, and so net ecosystem exchange (NEE) of CO2 remains quite stable across climatically diverse snow-free seasons. This is a valuable analysis of a fairly long EC data set, particularly for a tundra ecosystem. Overall I find the methodology to be quite sound and recommend several relatively minor but important revisions before the manuscript is considered further for publication. The following paragraphs describe more major issues, and are then followed by specific comments.*

We thank the reviewer for taking the time to assess our manuscript. We believe the comments have improved the manuscript.

*The introduction should be improved in several ways. First, the paragraph on flux partitioning seems out of place. The first and third paragraphs highlight research surrounding tundra/Arctic C cycling, and are bisected by the paragraph on partitioning. It would make more sense to first discuss carbon cycle dynamics and then highlight challenges associated with EC partitioning; so switch paragraphs two and three.*

The reviewer is correct that the paragraphs 2 and 3 should be inverted. The introduction has been modified based on the referee comment.

*In the results it seems that sections 3.3 should come before section 3.2; first describe the partitioning comparisons and then get into the results. Related, I don't see where you mention which partitioning/gapfilling methods you report. It would make sense to first present the flux processing results, and then state which date you'll present moving forward. Also, it is general good to have the figures ordered as they appear in the text. Currently order is Fig 5 -> Fig 4 -> Fig 3.*

The reviewer is correct that the sections 3.2 and 3.3 should be inverted. The results section has been improved. Now the partitioning/gapfilling method is presented before the results (L224-241).

Further, the figures have been ordered as they appear in the text.

*The last major area for revision is related to the broader implications of your results – specifically, how transferable are they? There is some of this in section 4.3, but it could be expanded there, and perhaps in section 4.1. Specifically, it occurs to me that this research site receives a relatively high amount of precipitation relative to many other tundra ecosystems, and has no permafrost. As such, the NEE responses to climate at other tundra sites may likely be more variable. It would be worth discussing this a bit further.*

Text has been revised and implemented to focus on the implications of our results (L335-342):

> This site likely receives more precipitation relative to many other tundra ecosystems, and has no permafrost, thus the NEE response to climate could be less variable. However, Kobbefjord is located in a costal area, so it is not surprising to receive high precipitation, and other ecosystems such as coastal blanket bogs (Lund et al., 2015) often receive even more precipitation, without a clear impact on the NEE sensitivity. On the other hand permafrost adds another layer of complexity to the C dynamics. Although some studies showed similarities of $CO_2$ fluxes in various northern wetland ecosystems with and without permafrost (Lund et al., 2015), permafrost has strong influence on the hydrology of peatlands, and therefore their topography and distribution of vegetation. Especially in the context of climate warming permafrost thaw can cause large changes to the ecosystems.

*Secondly, it is difficult to talk about ecosystem CO2 source/sink dynamics without some discussion of non-growing season processes. Papers by Zona et al and Commaine etal (very recently) indicate the importance of non-growing season C dynamics. Also, given the fact that you are using net sink timing to define the growing season, I wonder what effect previous growing season or previous winter conditions might have on your results? For example does a wet summer followed by a warm winter lead to high Reco the following year? There are very likely some interesting time-lag effects influencing the patterns you observe. Again, you allude to these processes, for example, by mentioning previous winter temperatures, but I think a more targeted and thoughtful discussion on temporal lags/dynamics would be useful. Actually, it would be helpful to report non-growing season climate data, and perhaps even analysis of these sorts of time lags. I do not think the latter is absolutely necessary, because this paper already contains a lot of information, but it could be informative either here or in another paper.*

Graph 2b has been included and the corresponding text in the results section has been revised to include meteorology from non-growing season, including preceding cold season (October to May) and warm season (June to September) (L208-212).

[Figure]

Figure 2: (a) Annual Temperature (°C) and precipitation (mm) anomalies of the analyzed years (2008-2015) compared to the 1866-2007 time series shown as empty circles (Cappelen, 2016), and (b) within the 2008-2015 period including annual (January to December), warm season (July to September) and cold season (October to May) averages.

Among the eight study years (figure 2b), the warm season (June to September) temperature and precipitation anomalies ranged from approx. -1°C (2011, 2013 and 2015) to +1.5°C (2010) and -96 mm (2011) to approx. +125 mm (2012 and 2013), respectively. The cold season (October to May) anomalies have shown a significant increase of both temperature and precipitation variability. 2010 was the warmest year while 2011 and 2015 were the coldest years.

Moreover, some text has been implemented in the discussion l368-379

The year 2010 had the warmest mean annual temperature (3.4 °C compared to the -0.4 °C 2008-2015 mean) and the warmest mean wintertime temperature (-2.7 °C compared to -6.79 °C 2008-2015 mean)(Figure 2a). These climatic conditions stimulated the thinnest (snow accumulation of 142 m compared to averaged 567 m)(Table 1)** and shortest-lasting snowpack. Consequently, 2010 had the longest growing season (85 days) and very high growing season C uptake (-70 g C/ m$^{-2}$). Increases in temperature can lead to high respiration rates during early winter (Commane et al., 2017; Zona et al., 2016), but also during the following summer (Lund et al., 2012), which is related to soil temperature and snow dynamics. Kobbefjord had in 2011 one of the coldest mean annual temperature and mean wintertime temperature (-1.7 and -6.1°C respectively), which created the thickest (snow accumulation of 995 m) and the longest-lasting snowpack, stimulating the shortest growing season (only 47 days). According to Lund et al. (2012), below thick snowpack soils will be insulated from reaching low temperature; and at the same time the snowpack will act as a lid by increasing diffusive resistance, preventing $R_{eco}$ from being released to the atmosphere. After snowmelts, $CO_2$ stored in soil and snow cavities will be released.

**Table 1 has been implemented with snow accumulation (instead of maximum snow depth).

Further, we understand the referee point about the importance of non-growing season climate implications. Winter fluxes are beyond the scope of this paper, since it is hard to analyse only eight-years dataset, but that an ongoing modelling effort will seek to address these issues. The referee comment will be a good point to address in this coming paper.

*(I will also note here that it seems odd to place the section on EC processing between to two sections discussing CO2 dynamics).*

The sections have been inverted accordingly.

*Minor edits:*

o *Lines 40-44: You should explicitly state that you are referring to soil C stocks – this doesn't come until the very end.*

   Now corrected.

o *Line 76: Why do you mention C a need for sites with C stocks if you don't present them in the paper?*

   Although it is highly interesting to measure C stocks in the field, the reviewer is correct that we don't present C stocks data in this paper. Therefore, we decided to remove this part.

o *Line 102: This line is a bit too informal; it's not Skip's map, it was a large collaborative effort. It would be more appropriate to report the class and the name of the map and the paper describing the map. Walker, D. et al. (2005), The Circumpolar Arctic vegetation map, Journal of Vegetation Science, 16(267-282).*
o *Lines 103-104: I don't understand this, what does it mean that the site 'went out of the Arctic zone'?*

   Both parts have been adjusted accordingly l 99-103:

   Kobbefjord belongs to the "Arctic Shrub Tundra" (bioclimate zone E) according to The Circumpolar Arctic Vegetation Map (Walker et al., 2005; CAVM Team, 2003). This map is based on the summer warmth index (SWI), which is the sum of the monthly mean temperature above 0 °C from May to September, and the southernmost bioclimatic zone E has the limits 20-35. In 2010 and 2012, climate conditions led the area to experience temperatures from warmer climatic zone (SWI ca. 36 and 35 respectively).

o *Line 142: What is Papale et al In Prep? Perhaps indicate that this is via personal communication as well, if that is the case.*

   Reference deleted.

o *Line 264: This is a very simplistic and incomplete view of the residence time of fixed C. I'm not sure you can say anything meaningful about C residence time*

*with discussing fluxes between pools and storage, which aren't really addressed in this manuscript.*

Net flux information alone is not enough to determine residence times, which depend on internal flows, dynamics and pool sizes. So we adjusted this text to remove the discussion around residence times.

○ *Line 279: This could be worded clearer; at first I thought you were saying the PAR values peak at 6am, which was confusing. Perhaps explicitly state that the predictive importance of PAR peaks at this time.*

Sentence adjusted accordingly, "coinciding with the predictive importance of PAR"

○ *Line 287: The model 'catching' something is perhaps a bit too colloquial. Better to state that it revealed or indicated a decline in the importance of PAR in 2011.*

The text has been changed to "the Random Forest analysis revealed a decrease of PAR's importance in 2011"

○ *Line 295: You can only say that NEE is insensitive to climate during the snow-free season.*

Sentence implemented.

○ *Line 300: 'NEE exchange' is redundant, just use NEE (here and elsewhere).*

Corrected.

○ *Line 330: Lots of typos here.*

Thanks for finding these two errors; now corrected.

○ *Figures 4 & 7: It would be good to include a legend indicating what the colors represent, in addition to the text description.*

The legend has been updated in both Figure 4 and 7. In Figure 4, the facets' labels on the right have been increased in size as well for readability purposes. Moreover, it has been modified the colours of air temperature and precipitations, as well as the direction of the facets on the right. Further, Figure 6 has been also harmonized colour wise with respect Figure 7.

---

## Author Comment (AC2) · 21 Jul 2017

**Referee #2**

Received and published: 29 June 2017

The article "Exchange of CO2 in Arctic tundra: impacts of meteorological variations and biological disturbance" by Lopez-Blanco and co-authors presents eight years of eddy covariance measurements from a tundra site in Greenland. The data set is rich and the authors apply current and appropriate methods in data analysis to derive gap-filled net carbon fluxes, as well as to partition these fluxes into the photosynthetic and respiration components. The authors attempt to analyze gap-filling procedures and use autochamber data towards these efforts. The undertaken analyses reveal valuable insight into the behavior of tundra carbon cycling in response to environmental variability from hourly to inter annual scales. Novel methods are applied to analyze the role of environmental drivers of C cycling as well as biological factors such as a pest outbreak. In general, the manuscript is a solid and valuable contribution. Greater attention to grammar, structure, and clarity will greatly improve the article. In some cases, additional justification for statements or references to literature are needed. The comments that follow provide suggestions for addressing these concerns before publication.

We are thankful for the reviewer's insightful comments that have improved the manuscript. We have carefully considered the reviewer's remarks and clarified our manuscript accordingly.

**General comments:**

There is too much repetition in portions of the manuscript (specific comments identify some of these sections), and efforts to reduce repetition will increase the readability of the paper.

Taking your advice into account, and also based on your guideline in one the very last specific comments (P10), the conclusions have been reduced to put the key findings in a more general context.

More attention is needed to grammar throughout the manuscript. Importantly please play close attention to the correct use of singular or plural nouns. Here are some examples where they should be switched (but please address on a case by case basis): Singular case instead: temperatures -> temperature, exchanges -> exchange, budgets -> budget, precipitations -> precipitation, references -> reference, evidences -> evidence Data: plural -> data are rare Capitalize Earth and Arctic when proper nouns

Thanks for finding these errors; now corrected.

With respect to figure 6, what causes the different direction (clockwise vs counterclockwise) in the hysteresis observed in 2010 vs 2012 vs 2013? It would be interesting to know the whether the causes for early versus late season decoupling of GPP and Reco are the same or different.

This is actually a very good comment. The data suggest that the clockwise 2012 hysteresis was due to greater gross C cycling (GPP and  $R_{eco}$ ) in June and July; while in 2010 and 2013 (counter-clockwise hysteresis), the higher gross C fluxes have been measured in August.

In the following figures it has been plotted (from left to right) June, July and August temperature and precipitation anomalies. We can observe warmer and drier (in June) and warmer and wetter (in July) conditions in 2012 (yellow), whereas 2010 has had warmer and wetter conditions in August (light blue). These differences could explain the different direction in the hysteresis observed.

Temperature (°C) and precipitation (mm) anomalies in June (a), July (b) and August (c) of the analyzed years (2008-2015).

**Specific comments:**

• Abstract: I find the use of meteorology and climate to be a bit conflicting here. Please ensure whether you mean meteorology or climate with reference your conclusions in this study.

The referee is right; we should keep consistency in the terminology. Since this is an 8-years dataset, we decided to use the term meteorology rather than climate.

• *P2L69: The terminology "C balance state" does not carry an immediate clear meaning.*

Does this refer to the annual balance of net carbon exchange? Clarify what C state refers to and how relates to fluxes versus carbon stocks and over which time frames. What is your definition of C uptake and C storage, and over what time frame?

The text has been changed to sign and magnitude of the C balance instead

• *P2L52: Eddy covariance data can include other types of gases, so good to specify: Eddy covariance measurements of CO2*

Corrected accordingly

• *P3L82: Resiliency in which sense? Should clarify right away.*

We meant the resiliency of the sink. However, this part has been removed, so the objectives are more direct and clear. The resiliency of the sink will be just briefly mentioned in the discussion.

• P3: Sections of the end of the introduction are too detailed to be placed in the introduction and should be moved to the materials and methods section. Please separate material between L82-91 into intro vs methods as appropriate

Following your suggestion, we have moved the second part of the paragraph into section 2.2 (Measurements) (L109-113).

 $\circ$  *P3L116: clarify what* 5+5 *min means*

The computer running these automatic measurements activates the chambers in succession for 10 minutes. During the first 3 minutes the chamber is open for ventilation, then closed for 5 minutes, and opened again for the last 2 minutes. Each chamber is therefore activated once per cycle while the inactive chambers remain open.

In the text we have updated "in succession for 10 min every hour" (L120)

• P3L120: spell out km if used in this sentence

Corrected accordingly

• *P5L184: Please clarify what is meant by "sums the variable's importance up to 1". This sentence could be clearer*

We changed the sentence to: "This version of Random Forest sums the relative importance of each variable from 0% up to 100 %, which correspond to the fraction of decision in which a variable is involved to cluster the data." (L188-190)

o P5L198: Check grammar: "also exposed a larger variability"

Corrected: "also exhibited larger variability".

• *P6L205: what is a non-lap year?*

*Typo, we meant non-leap year. Now corrected.*

• *P6L216: measurement period*

Agreed, changed accordingly.

• P6L223 & Fig S4: The largest GPP and Reco were found in wetter and warmer years, but what is the statistical measure to support a "tendency towards larger GPP and Reco during wetter and warmer years"? For example, for Reco, half of warmer/wetter are larger and half are smaller than colder/drier.

The referee is right, the Figure S4 (now Figure S3) is not correct as such. It shows the annual and precipitation anomaly of the analyzed years (2008-2015) compared to the 1866-2007 time series. This graph should only include the anomalies within the measurement period (i.e. 2008-2015). Based on Ref#1, we have updated the Figure 2b including annual, cold and warm periods during the measurement period. This new input shows that both annual and cold period in 2010, 2012 and 2013 had larger GPP and Reco during wetter and warmer conditions. Figure S3 has been also updated.

Figure S3: Annual cumulative GPP and  $R_{eco}$  defined by annual temperature and precipitation anomalies (2008-2015). The flux size is categorized depending on the flux magnitude (g C m-2), i.e. larger diameters with greater fluxes.

• P6L228: perhaps be more specific about what the response to the outbreak was in terms of fluxes (not really a response of measurements, but of actual fluxes). Just GPP?

We have implemented the text to: "coinciding with high NEE and very low GPP (Figure 4)" (L258).

o P7L281: I wouldn't use "momentarily" to describe hourly data

We wrote "temporarily" instead.

• P7L285: What is meant by "although Tair appeared to be the less limiting factor". It seems that Tair is the most important variable for Reco, but I'm not sure how it would be limiting or not

The referee is correct, the sentence as such sounds odd. We reformulated the text: In terms of  $CO_2$  release ( $R_{eco}$ ) the pattern is less clear and noisier, although  $T_{air}$  appeared to be the most important variable.

• P8L1286: Check grammar in the last sentence. I wouldn't use "catch". Please elaborate on what the connection here is. Why would a decrease in PAR's importance are sense here?

We changed "catches" with "revealed". PAR is interesting because it includes information about cloudiness. Negative PAR anomalies in 2011 show less bright growing season compared to the other years, which could have contributed to the C dynamics in the cited year.

• P8L293: What tendency is that? Also, don't use 'mirror effect'. Use clearer language.

The first part of this point has been answer earlier (P6L223 & Fig S4). With respect the mirror effect wording, we updated the text:

"The results suggest that the relative insensitivity of NEE to meteorological conditions during the snow-free period could be the result of the correlated response of ranked cumulative GPP and  $R_{eco}$  (Figure 5). In this study, larger rates of C uptake (GPP) implied also larger rates of C release ( $R_{eco}$ ), with exception of the anomalous year 2011." (L333-336)

• P8L298: I'm not sure this sentence is a natural conclusion from your results: "Thus, the effects on C balance of warming from climate change are not straightforward to infer." Would these processes not be predicted by models? If so, then it could be misleading to state that it is difficult to infer. Provide some context from current literature here if in fact current understanding would have missed this.

We implemented the text with the following explanation:

Further, this study agrees with Parmentier et al., (2011), who suggested that a longer growing season does not necessarily increase the net carbon uptake. Here a more negative NEE indicated a stronger C sink (i.e.) in 2012 compared to 2010. Parmentier et al., (2011) hypothesized that this behavior is due to site-specific differences, such as meteorology and soil structure, and that changes in the carbon cycle with longer growing seasons will not be uniform around the Arctic. Thus, the effects of climate change on the tundra C balance of are not straightforward to infer. (L344-348)

• *P8L303: a bit redundant with 'growing season' twice*

Agreed, we took out the first "growing season".

• *P8L314: outbreak of what?*

We updated the text with "outbreak of autumn and winter moths". (L363)

o P8L317, L330: check grammar

Corrected, previous referee also pointed towards this sentence. Thanks.

• P8L322: shortest-lasting, longest-lasting

Corrected

• *P9L337: This first two sentences are very unclear as written*

Agreed, we updated the text in a clearer way.

"The NEE gap-filling and subsequent partitioning are needed to understand the responses to the environmental forcing. However, these procedures expose partial inconsistencies between approaches (Figure 4) and unavoidable uncertainties in the seasonal C budget calculation (Table 2)." (L302-304)

• P9 section 4.2: I don't find this analysis of gap filling to be very informative because estimates regarding which method is best are not testable. Why not test the performance of the gap-filling on years where you have good data coverage by creating artificial gaps and testing model performance against real data? I would find that exercise to be much more compelling and would help you determine which method to apply in years where data is really missing.

Quantifying the uncertainty introduced by measurement gaps is difficult. One possibility would be a sensitivity analysis of time series with artificially introduced gaps as the referee suggest. But the choice of gap length and position is difficult, and would render the uncertainty assessment itself quite uncertain. We think the paper already contains a lot of information. So the best way to give the reader an idea about why we decided to use the auto-chamber (AC) data is that the MDS gap-filling alone introduced NEE values out of range. Instead of blindly trust a gap-filling script, which create odd numbers, we decreased the gap length introducing AC data. We understand AC data incorporated uncertainties to the calculations, although they have been included in the total uncertainty estimation.

• *P9L365: How was the filtering done? This is not clear.*

We separated the dataset in 3 subgroups: all day data (0-24hr), daytime data (11-14hr) [when GPP is the strongest and will represent the largest part of NEE], and nighttime data (00-03hr) [when NEE= $R_{eco}$ ]. By doing this, we make sure that the Random Forest approach will not include bias from the partitioning analysis.

P10: I would avoid using 'interesting' so much as a way to describe your observations. It would be more informative to put in context with extant literature. You should not just repeat results here that are listed elsewhere, but put into context. For example, this is done in the latter half of the L380-387 paragraph, but not the first part. The first half of the conclusion is a bit repetitive as well - should not be a repetition of abstract, should be more general.

Two 'interesting' words have been removed from the text.

The conclusion part was reduced to put the key findings in a more general context, omitting detailed values and districting information which has been addressed previously in the results and discussion sections (L417-424):

We have analyzed eight snow-free periods in eight consecutive years in a West Greenland tundra (64° N) focusing on the net ecosystem exchange (NEE) of CO2 and its photosynthetic inputs (GPP) and respiration outputs ( $R_{eco}$ ). We find that Kobbefjord acted as a consistent sink of CO2, during the years 2008-2015, except 2011 that was associated with a major pest outbreak. The results do not show a marked meteorological effect on the net C uptake. The relative insensitivity of NEE during the snow-free period was the result of the correlated response of GPP and  $R_{eco}$ . The ranges in annual GPP and  $R_{eco}$  were >5 fold larger and more variable than for NEE. Here we show a tendency towards larger GPP and  $R_{eco}$  during wetter and warmer years. The anomalous year, 2011, constituted a relatively strong source for CO2 and has decreased its C sink strength due to the biological disturbance, which reduced GPP more strongly than  $R_{eco}$ . The changes of environmental forcing across diurnal, seasonal and annual time scales unmasked patterns of functional responses to C fluxes.

• Table S1: Avoid using N•

Corrected.

• Where is Figure S1?

The indexing in the supplementary material has been changed to Equations S1, Figures S1, S2, S3 and S4, Tables S1 and S2.

---

## Author Response (AR1)

**Referee #1**

**Lopez-Blanco and colleagues present a study of ecosystem CO2 dynamics across eight snow-free seasons for a wet fen tundra ecosystem in west Greenland. The authors compare ecosystem respiration (Reco) and gross primary production (GPP) with key climatic drivers to characterizes how ecosystem CO2 dynamics will change with climate. Comparisons are made at hourly, daily, and seasonal timescales to understand how drivers of ecosystem CO2 dynamics change across temporal scales. Additionally, the authors compare several eddy covariance partitioning methods in order to assess uncertainty associated with interpretation of EC derived estimates of Reco and GPP. The main finding is that large interannual variations in Reco and GPP with climate are compensatory, and so net ecosystem exchange (NEE) of CO2 remains quite stable across climatically diverse snow-free seasons. This is a valuable analysis of a fairly long EC data set, particularly for a tundra ecosystem. Overall I find the methodology to be quite sound and recommend several relatively minor but important revisions before the manuscript is considered further for publication. The following paragraphs describe more major issues, and are then followed by specific comments.**

We thank the reviewer for taking the time to assess our manuscript. We believe the comments have improved the manuscript. We carefully considered each of the comments, paying special attention to the structure of the paper and the implications and the transferability of our findings.

**The introduction should be improved in several ways. First, the paragraph on flux partitioning seems out of place. The first and third paragraphs highlight research surrounding tundra/Arctic C cycling, and are bisected by the paragraph on partitioning. It would make more sense to first discuss carbon cycle dynamics and then highlight challenges associated with EC partitioning; so switch paragraphs two and three.**

The reviewer is correct that the paragraphs 2 and 3 should be inverted. The introduction has been modified based on the referee comment (L47-63). Moreover, we moved the information about the measurements to the materials and methods section (L105-109). Further, we included our overarching hypothesis at the end of the introduction (L83-85).

**In the results it seems that sections 3.3 should come before section 3.2; first describe the partitioning comparisons and then get into the results. Related, I don't see where you mention which partitioning/gapfilling methods you report. It would make sense to first present the flux processing results, and then state**

**which date you'll present moving forward. Also, it is general good to have the figures ordered as they appear in the text. Currently order is Fig 5 -> Fig 4 -> Fig 3.**

The reviewer is correct that the sections 3.2 and 3.3 should be inverted. The results section has been improved. Now the partitioning/gapfilling method is presented before the results (L219-238).

Further, the figures have been ordered as they appear in the text.

**The last major area for revision is related to the broader implications of your results – specifically, how transferable are they? There is some of this in section 4.3, but it could be expanded there, and perhaps in section 4.1. Specifically, it occurs to me that this research site receives a relatively high amount of precipitation relative to many other tundra ecosystems, and has no permafrost. As such, the NEE responses to climate at other tundra sites may likely be more variable. It would be worth discussing this a bit further.**

Text has been revised and implemented to focus on the implications of our results (L332-347):

Interestingly, the tendency to warmer and wetter conditions led to greater rates of C cycling associated with larger GPP and $R_{eco}$ (Figure S3; supplementary material). This result does not entirely coincide with Peichl et al. (2014), even though they performed a similar analysis for a Swedish boreal fen. This finding points towards the complexity in the response of wetland ecosystems towards changing environmental conditions. The response is dependent on many things, such as hydrological settings, and these differ between sites. In this study, larger rates of C uptake (GPP) were linked to larger rates of C release ($R_{eco}$), with the exception of the anomalous year 2011. The relative insensitivity of NEE to meteorological conditions during the snow-free period could be the result of the correlated response of ranked cumulative GPP and $R_{eco}$ (Figure 5) (Richardson et al., 2007; Wohlfahrt et al., 2008). This site likely receives more precipitation relative to many other tundra ecosystems, and has no permafrost, thus the NEE response to climate could be less variable. However, as Kobbefjord is located in a coastal area, it is not surprising to receive high precipitation, and other ecosystems such as coastal blanket bogs often receive even more precipitation without a clear impact of drought effect on the NEE sensitivity (Lund et al., 2015). Furthermore, permafrost adds another layer of complexity to the C dynamics (Christensen et al., 2004; Koven et al., 2011; Schuur et al., 2015). Although some studies showed similarities of $CO_2$ fluxes in various northern wetland ecosystems with and without permafrost (Lund et al., 2015), permafrost has strong influence on the hydrology of peatlands (Åkerman and Johansson, 2008), and therefore their topography and distribution of vegetation (Johansson et al., 2013). Especially in the context of climate warming permafrost thaw can cause large changes to the ecosystems.

**Secondly, it is difficult to talk about ecosystem CO2 source/sink dynamics without some discussion of non-growing season processes. Papers by Zona et al and Commaine etal (very recently) indicate the importance of non-growing season C dynamics. Also, given the fact that you are using net sink timing to define the growing season, I wonder what effect previous growing season or**

**previous winter conditions might have on your results? For example does a wet summer followed by a warm winter lead to high Reco the following year? There are very likely some interesting time-lag effects influencing the patterns you observe. Again, you allude to these processes, for example, by mentioning previous winter temperatures, but I think a more targeted and thoughtful discussion on temporal lags/dynamics would be useful. Actually, it would be helpful to report non-growing season climate data, and perhaps even analysis of these sorts of time lags. I do not think the latter is absolutely necessary, because this paper already contains a lot of information, but it could be informative either here or in another paper.**

We have adjusted Figure 2 and the corresponding text in the results section to include meteorology from non-growing season, including preceding cold season (October to May) and warm season (June to September) (L720-725).

[Figure]

Figure 2: (a) Annual Temperature (°C) and precipitation (mm) anomalies of the analyzed years (2008-2015) compared to the 1866-2007 time series shown as empty circles (Cappelen, 2016), and (b) within the 2008-2015 period including annual (January to December), warm season (July to September) and cold season (October to May) averages.

Moreover, we supported the Figure with more text (L206-210):

Among the eight study years (figure 2b), the warm season (June to September) temperature and precipitation anomalies ranged from approx. -1°C (2011, 2013 and 2015) to +1.5°C (2010) and -96 mm (2011) to approx. +125 mm (2012 and 2013), respectively. The cold season (October to May) anomalies have shown a significant increase of both temperature and precipitation variability. 2010 was the warmest year while 2011 and 2015 were the coldest years.

Further, some text has been implemented in the discussion l372-383:

A combination of different factors could have led to the sharp change in C balance observed between 2010-2011, both physical and biological. The year 2010 had the warmest mean annual temperature (3.4 °C compared to the -0.4 °C mean annual

temperature for 2008-2015) and the warmest mean wintertime temperature (-2.7 °C compared to the -6.79 °C mean for 2008-2015) (Figure 2a). These climatic conditions generated the thinnest (maximum daily snow depth of 0.3 m compared to averaged 0.9 m) (Table 1) and shortest-lasting snowpack. Consequently, 2010 had the longest growing season (85 days) and very high growing season C uptake (-70 g C/ $m^{-2}$). Increases in temperature can lead to high respiration rates during early winter (Commane et al., 2017; Zona et al., 2016), but also during the following summer (Helfter et al., 2015; Lund et al., 2012), which is related to soil temperature and snow dynamics. Further, in Kobbefjord the year 2011 had one of the lowest mean annual temperatures and mean wintertime temperatures (-1.7 and -6.1°C respectively), which created the thickest (maximum daily snow depth of 1.4 m) and the longest-lasting snowpack, leading to the shortest growing season for the study period (only 47 days). According to Lund et al. (2012), below thick snowpack soils will be insulated from reaching low temperature, acting as lid and preventing $R_{eco}$ from being released to the atmosphere until the snowmelt period.

Finally, we understand the referee's point about the importance of non-growing season climate implications. Winter fluxes are beyond the scope of this paper, since it is hard to analyse only eight-years dataset, but that an ongoing modelling effort will seek to address these issues. The referee comment will be a good point to address in this coming paper.

**(I will also note here that it seems odd to place the section on EC processing between to two sections discussing CO2 dynamics).**

The sections have been inverted accordingly.

**Minor edits:**

o **Lines 40-44: You should explicitly state that you are referring to soil C stocks – this doesn't come until the very end.**

  Now corrected.

o **Line 76: Why do you mention C a need for sites with C stocks if you don't present them in the paper?**

  Although it is highly interesting to measure C stocks in the field, the reviewer is correct that we don't present C stocks data in this paper. Therefore, we decided to remove this part.

o **Line 102: This line is a bit too informal; it's not Skip's map, it was a large collaborative effort. It would be more appropriate to report the class and the name of the map and the paper describing the map. Walker, D. et al. (2005), The Circumpolar Arctic vegetation map, Journal of Vegetation Science, 16(267-282).**
o **Lines 103-104: I don't understand this, what does it mean that the site 'went out of the Arctic zone'?**

  Both parts have been adjusted accordingly L96-100:

Kobbefjord belongs to the "Arctic Shrub Tundra" (bioclimate zone E) according to The Circumpolar Arctic Vegetation Map (CAVM Team, 2003; Walker et al., 2005). This map is based on the summer warmth index (SWI), which is the sum of the monthly mean temperature above 0 °C from May to September and the southernmost bioclimatic zone E has the limits 20-35. In 2010 and 2012, climate conditions led the area to experience temperatures from warmer climatic zones (SWI ca. 36 and 35 respectively).

o **Line 142: What is Papale et al In Prep? Perhaps indicate that this is via personal communication as well, if that is the case.**

Reference deleted.

o **Line 264: This is a very simplistic and incomplete view of the residence time of fixed C. I'm not sure you can say anything meaningful about C residence time with discussing fluxes between pools and storage, which aren't really addressed in this manuscript.**

Net flux information alone is not enough to determine residence times, which depend on internal flows, dynamics and pool sizes. So we adjusted this text to remove the discussion around residence times.

o **Line 279: This could be worded clearer; at first I thought you were saying the PAR values peak at 6am, which was confusing. Perhaps explicitly state that the predictive importance of PAR peaks at this time.**

Sentence adjusted accordingly (L281): "*PAR was important at dawn (06 h. WGST) and dusk (15-17 h. WGST), while $T_{air}$ was more important at other times*".

o **Line 287: The model 'catching' something is perhaps a bit too colloquial. Better to state that it revealed or indicated a decline in the importance of PAR in 2011.**

The text has been changed (L292) to "the Random Forest analysis revealed a decrease of PAR's importance in 2011".

o **Line 295: You can only say that NEE is insensitive to climate during the snow-free season.**

Sentence implemented.

o **Line 300: 'NEE exchange' is redundant, just use NEE (here and elsewhere).**

Corrected.

o **Line 330: Lots of typos here.**

Thanks for finding these two errors; now corrected.

o **Figures 4 & 7: It would be good to include a legend indicating what the colors represent, in addition to the text description.**

The legend has been updated in both Figure 4 (L731) and 7 (L742). In Figure 4, the facets' labels on the right have been increased in size as well for readability purposes. Moreover, it has been modified the colours of air temperature and precipitations, as well as the direction of the facets on the right. Further, Figure 8 (L748) has been also harmonized colour wise with respect Figure 7.
* * ** * *
**The article "Exchange of CO2 in Arctic tundra: impacts of meteorological variations and biological disturbance" by Lopez-Blanco and co-authors presents eight years of eddy covariance measurements from a tundra site in Greenland. The data set is rich and the authors apply current and appropriate methods in data analysis to derive gap-filled net carbon fluxes, as well as to partition these fluxes into the photosynthetic and respiration components. The authors attempt to analyze gap-filling procedures and use autochamber data towards these efforts. The undertaken analyses reveal valuable insight into the behavior of tundra carbon cycling in response to environmental variability from hourly to inter annual scales. Novel methods are applied to analyze the role of environmental drivers of C cycling as well as biological factors such as a pest outbreak. In general, the manuscript is a solid and valuable contribution. Greater attention to grammar, structure, and clarity will greatly improve the article. In some cases, additional justification for statements or references to literature are needed. The comments that follow provide suggestions for addressing these concerns before publication.**

We are thankful for the reviewer's insightful comments that have improved the manuscript. We have carefully considered the reviewer's remarks and clarified our manuscript accordingly.

**General comments:**
**There is too much repetition in portions of the manuscript (specific comments identify some of these sections), and efforts to reduce repetition will increase the readability of the paper.**

Taking your advice into account, and also based on your guideline in one the very last specific comments (P10), major efforts have been dedicated to the discussion section, so the outcome is more transferable to literature and less repetitive. It was a priority to improve clarity and readability. We also worked in the paper's structure. Finally, the conclusions have been reduced to put the key findings in a more general context.

**More attention is needed to grammar throughout the manuscript. Importantly please play close attention to the correct use of singular or plural nouns. Here are some examples where they should be switched (but please address on a case by case basis): Singular case instead: temperatures -> temperature, exchanges -> exchange, bud- gets -> budget, precipitations -> precipitation, references ->**

**reference, evidences -> evidence Data: plural -> data are rare Capitalize Earth and Arctic when proper nouns**

Thanks for finding these errors; now corrected.

**With respect to figure 6, what causes the different direction (clockwise vs counter-clockwise) in the hysteresis observed in 2010 vs 2012 vs 2013? It would be interesting to know the whether the causes for early versus late season decoupling of GPP and Reco are the same or different.**

This is a very pertinent comment. We implemented the following figure in the supplementary material:

[Figure]

Figure S4: Temperature (°C) and precipitation (mm) anomalies in June, July and August of the analysed years (2008-2015).

Moreover, we implemented the following text in the results section (L269-273):

It is worth mentioning the different direction (clockwise vs counter-clockwise) in the hysteresis observed these years between June, July and August. The data suggest that the clockwise 2012 hysteresis was due to greater gross C cycling (GPP and $R_{eco}$) in June and July favored by warmer conditions; while in 2010 (counter-clockwise hysteresis), the higher gross C fluxes have been measured in August with warmer and wetter conditions (Figure S4; supplementary material).

*Specific comments:*

o **Abstract: I find the use of meteorology and climate to be a bit conflicting here. Please ensure whether you mean meteorology or climate with reference your conclusions in this study.**

The referee is right; we now maintain consistency in the terminology. Because this is an 8-years dataset, we have decided to use the term meteorology rather than climate.

- **P2L69: The terminology "C balance state" does not carry an immediate clear meaning. Does this refer to the annual balance of net carbon exchange? Clarify what C state refers to and how relates to fluxes versus carbon stocks and over which time frames. What is your definition of C uptake and C storage, and over what time frame?**

  The text has been changed to sign and magnitude of the C balance instead.

- **P2L52: Eddy covariance data can include other types of gases, so good to specify: Eddy covariance measurements of CO2**

  Corrected accordingly.

- **P3L82: Resiliency in which sense? Should clarify right away.**

  We meant the resiliency of the sink. However, this part has been removed, so the objectives are more direct and clear. The resiliency of the sink will just be briefly mentioned and defined in the discussion.

- **P3: Sections of the end of the introduction are too detailed to be placed in the introduction and should be moved to the materials and methods section. Please separate material between L82-91 into intro vs methods as appropriate**

  Following your suggestion, we have moved the second part of this paragraph into section 2.2 (Measurements) (L106-110).

- **P3L116: clarify what 5+5 min means**

  The computer running these automatic measurements activates the chambers in succession for 10 minutes. During the first 3 minutes the chamber is open for ventilation, then closed for 5 minutes, and opened again for the last 2 minutes. Each chamber is therefore activated once per cycle while the inactive chambers remain open.

  In the text we have updated *"in succession for 10 min every hour"* (L117)

- **P3L120: spell out km if used in this sentence**

  Corrected accordingly.

- **P5L184: Please clarify what is meant by "sums the variable's importance up to 1". This sentence could be clearer**

  We changed the sentence to: "This version of Random Forest sums the relative importance of each variable from 0% up to 100 %, which correspond to the fraction of decision in which a variable is involved to cluster the data." (L186-187)

- **P5L198: Check grammar: "also exposed a larger variability"**

Corrected: "also exhibited larger variability".

o **P6L205: what is a non-lap year?**

*Typo, we meant non-leap year. Now corrected.*

o **P6L216: measurement period**

Agreed, changed accordingly.

o **P6L223 & Fig S4: The largest GPP and Reco were found in wetter and warmer years, but what is the statistical measure to support a "tendency towards larger GPP and Reco during wetter and warmer years"? For example, for Reco, half of warmer/wetter are larger and half are smaller than colder/drier.**

The referee is right, the Figure S4 (now Figure S3) is not correct as such. It shows the annual and precipitation anomaly of the analyzed years (2008-2015) compared to the 1866-2007 time series. This graph should only include the anomalies within the measurement period (i.e. 2008-2015). Based on Ref#1, we have updated Figure 2b to include annual, cold and warm periods during the measurement period. This new input shows that 2010, 2012 and 2013 were relatively warmer and wetter, and had larger GPP and $R_{eco}$. Figure S3 has been also updated.

[Figure]

Figure S3: Annual cumulative GPP and $R_{eco}$ defined by annual temperature and precipitation anomalies (2008-2015). The flux size is categorized depending on the flux magnitude (g C m$^{-2}$), i.e. larger diameters with greater fluxes.

o **P6L228: perhaps be more specific about what the response to the outbreak was in terms of fluxes (not really a response of measurements, but of actual fluxes). Just GPP?**

We have implemented the text to: "coinciding with high NEE and very low GPP (Figure 4)." (L258)

- o **P7L281: I wouldn't use "momentarily" to describe hourly data**

  This sentence was changed completely (L286): "*PAR was important at dawn (06 h. WGST) and dusk (15-17 h. WGST), while $T_{air}$ was more important at other times*".

- o **P7L285: What is meant by "although Tair appeared to be the less limiting factor". It seems that Tair is the most important variable for Reco, but I'm not sure how it would be limiting or not**

  The referee is correct, the sentence as such sounds odd. We reformulated the text (L285-286): *In terms of $CO_2$ emission ($R_{eco}$) the pattern is less clear and noisier, although $T_{air}$ appeared to be the most important variable.*

- o **P8L1286: Check grammar in the last sentence. I wouldn't use "catch". Please elaborate on what the connection here is. Why would a decrease in PAR's importance are sense here?**

  We changed "catches" with "revealed". PAR is interesting because it includes information about cloudiness. Negative PAR anomalies in 2011 show less bright growing season compared to the other years, which could have contributed to the C dynamics in the cited year.

- o **P8L293: What tendency is that? Also, don't use 'mirror effect'. Use clearer language.**

  The first part of this point has been answer earlier (P6L223 & Fig S4). With respect the mirror effect wording, we updated the text (L337-340*)*:

  In this study, larger rates of C uptake (GPP) were linked to larger rates of C release ($R_{eco}$), with the exception of the anomalous year 2011. The relative insensitivity of NEE to meteorological conditions during the snow-free period could be the result of the correlated response of ranked cumulative GPP and $R_{eco}$ (Figure 5) (Richardson et al., 2007; Wohlfahrt et al., 2008)

- o **P8L298: I'm not sure this sentence is a natural conclusion from your results: "Thus, the effects on C balance of warming from climate change are not straightforward to infer." Would these processes not be predicted by models? If so, then it could be misleading to state that it is difficult to infer. Provide some context from current literature here if in fact current understanding would have missed this.**

  We believe that models would not necessarily predict these results. For example, the results suggest that autotrophic respiration (Ra) and heterotrophic respiration (Rh) respond to climate together to balance GPP changes. But Ra and Rh are different processes operating on different pools, so such convergence is unlikely

in models.

We implemented the text, providing some context from literature (L349-353):

> Further, this study agrees with Parmentier et al., (2011) and Lund et al., (2012), who suggested that a longer growing season does not necessarily increase the net carbon uptake. Here a more negative NEE indicated a stronger C sink (i.e.) in 2012 compared to 2010. Parmentier et al., (2011) hypothesized that this behavior is due to site-specific differences, such as meteorology and soil structure, and that changes in the carbon cycle with longer growing seasons will not be uniform around the Arctic. Thus, the effects of climate change on the tundra C balance of are not straightforward to infer.

o **P8L303: a bit redundant with 'growing season' twice**

Agreed, we took out the first "growing season".

o **P8L314: outbreak of what?**

We updated the text with "outbreak of autumn and winter moths". (L368)

o **P8L317, L330: check grammar**

Corrected, previous referee also pointed towards this sentence. Thanks.

o **P8L322: shortest-lasting, longest-lasting**

Corrected

o **P9L337: This first two sentences are very unclear as written**

Agreed, we updated the text in a clearer way (L297-299):

> The NEE gap-filling and subsequent partitioning into GPP and $R_{eco}$ are needed to understand the $CO_2$ flux responses to the environmental forcing. However, these procedures expose unavoidable uncertainties in the seasonal C budget calculation (Table 2) and partial inconsistencies between approaches (Figure 4).

o **P9 section 4.2: I don't find this analysis of gap filling to be very informative because estimates regarding which method is best are not testable. Why not test the performance of the gap-filling on years where you have good data coverage by creating artificial gaps and testing model performance against real data? I would find that exercise to be much more compelling and would help you determine which method to apply in years where data is really missing.**

Quantifying the uncertainty introduced by measurement gaps is difficult. One possibility would be a sensitivity analysis of time series with artificially introduced gaps as the referee suggest. But the choice of gap length and position is

difficult, and would render the uncertainty assessment itself quite uncertain. We think the paper already contains a lot of information and this extra analysis would broaden its scope still further. So the best way to give the reader an idea about why we decided to use the auto-chamber (AC) data is that the MDS gap-filling alone introduced NEE values out of range. Instead of blindly trust a gap-filling script, which create odd numbers, we decreased the gap length introducing AC data. We understand AC data incorporated uncertainties to the calculations, although they have been included in the total uncertainty estimation.

We incorporated this discussion into the manuscript (L304-312):

> Quantifying the uncertainty introduced by measurement gaps is complex (Falge et al., 2001; Moffat et al., 2007; Papale et al., 2006). One possibility would be a sensitivity analysis of time series with artificially introduced gaps (Dragomir et al., 2012; Pirk et al., 2017). But the choice of gap length and position is difficult, and would render uncertainty to the uncertainty assessment itself. Instead, we used the EC prediction based on independent auto-chamber (AC) measurements between 2010 and 2013. The agreement between EC and AC were always $R^2 > 0.72$ and p <0.001, and the 95% confidence interval of the predictions were reported together with the resulting uncertainties (Table 2). Although the AC data itself incorporated a new source of uncertainty to the calculations, we consider this method to be less weak than an unreliable gap-filling estimate. We used the AC as platform to decrease the gap length and the total random uncertainty (Aurela et al., 2002) before the MDS algorithm was applied. AC was used together with MDS, and never was used as an independent gap-filling procedure.

o **P9L365: How was the filtering done? This is not clear.**

We separated the dataset in 3 subgroups: all day data (0-24hr), daytime data (11-14hr) [when GPP is the strongest and will represent the largest part of NEE], and nighttime data (00-03hr) [when NEE=$R_{eco}$]. By doing this, we make sure that the Random Forest approach will not include bias from the partitioning analysis.

o **P10: I would avoid using 'interesting' so much as a way to describe your observations. It would be more informative to put in context with extant literature. You should not just repeat results here that are listed elsewhere, but put into context. For example, this is done in the latter half of the L380-387 paragraph, but not the first part. The first half of the conclusion is a bit repetitive as well - should not be a repetition of abstract, should be more general.**

Two 'interesting' words have been removed from the text.

Following the referee's guideliness, we have worked in the discussion section, not only in the cited areas, but also across the previous subsections.

Moreover, the conclusion part was reduced to put the key findings in a more general context, omitting detailed values and information that has been addressed previously in the results and discussion sections (L428-435):

We have analyzed eight snow-free periods in eight consecutive years in a West Greenland tundra (64° N) focusing on the net ecosystem exchange (NEE) of $CO_2$ and its photosynthetic inputs (GPP) and respiration outputs ($R_{eco}$). Here, the NEE gap-filling exposed inherent uncertainties in the seasonal C budget calculation, but there were also inconsistencies between the flux partitioning approaches used. We find that Kobbefjord acted as a consistent sink of $CO_2$, during the years 2008-2015, except 2011 that was associated with a major pest outbreak. The results do not show a marked meteorological effect on the net C uptake. However, the relative insensitivity of NEE during the snow-free period was driven by the correlated, balancing responses of GPP and $R_{eco}$, both more variable than NEE and sensitive to temperature and insolation. In this paper we show a tendency towards larger GPP and $R_{eco}$ during wetter and warmer years. The anomalous year 2011, affected by a biological disturbance, constituted a relatively strong source for $CO_2$ and reduced GPP more strongly than $R_{eco}$. A novel analysis assessing the changes of environmental forcing across diurnal, seasonal and annual time scales unmasked patterns of functional responses to C fluxes.

o **Table S1: Avoid using N∘**

Corrected.

o **Where is Figure S1?**

The indexing in the supplementary material has been changed to Equations S1, Figures S1, S2, S3 and S4, Tables S1 and S2.
* * *

[revised manuscript text omitted]
 present different C exchange rates (Bubier et al., 2003), and because the composition of vegetation varies as a response to environmental changes (Glenn et al., 2006), C exchange presents correlated responses. Synthesis studies have found a significant spatial variability in NEE (Lafleur et al., 2012; Mbufong et al., 2014) between different sites in the Arctic tundra (Lindroth et al., 2007; Lund et al., 2010) but also a large temporal variability within sites (Aurela et al., 2004; Aurela et al., 2007; Christensen et al., 2012; Grøndahl et al., 2008; Lafleur et al., 2012). Minor variations in GPP and $R_{eco}$ may promote changes in the C balance state (Arndal et al., 2009; Elberling et al., 2008; IPCC, 2007; Lund et al., 2010; Tagesson et al., 2012; Williams et al., 2000). With continued warming temperatures and longer growing seasons, tundra systems will likely have enhanced GPP and $R_{eco}$ rates, but long-term data to investigate these responses is rare. Further, the effects on net $CO_2$ sequestration are not known, and may be altered by long-term processes such as vegetation shifts and short-term disturbances like insect pest outbreaks, complicating the prognostic forecast of upcoming C states (Callaghan et al., 2012b; McGuire et al., 2012). Consequently, there is a need to understand how C cycle behaves over time scales from days to years, and the links to environmental drivers. There is a lack of reference sites from where full measurement-based data is available, documenting the basic carbon stocks and fluxes at the terrestrial catchment scales. Here we investigate the functional responses of C exchange to environmental characteristics across eight snow-free periods in eight consecutive years in West Greenland.

The main objectives of this paper are (1) to explore the uncertainties in NEE gap-filling and partitioning obtained from different approaches, (2) to determine how C uptake and C storage respond to the meteorological variability, and assess the resiliency of the studied ecosystem to meteorological variability, and (3) to identify how the environmental forcing affects not only the inter-annual variability, but also the hourly, daily, weekly and monthly variability of NEE, GPP and $R_{eco}$. The intention of this paper is to elaborate on the information gathered in an existing catchment area under an extensive cross-disciplinary ecological monitoring program in low Arctic West Greenland, established under the auspices of the Greenland Ecosystem Monitoring (GEM) (http://www.g-e-m.dk). Using a long-term (8 years) dataset to explore uncertainties in NEE gap-filling and partitioning methods and to characterise the inter-annual variability of C exchange in relation to driving factors can provide a novel input into our understanding of land-atmosphere $CO_2$ exchange in Arctic regions. Our overarching hypothesis was that both GPP and $R_{eco}$ would respond positively to warmer and longer growing seasons; but, that NEE response to warming would be more complex and variable (positive or negative), depending on subtle balances between plant and microbial climate sensitivity. The time series is focused on the snow free period, our measurements typically start around the end of the snow melt (ca. May-June) and extend 
[revised manuscript text omitted]
 computationcalculation. During the eight study snow-free periodsyears growing seasons, there weredata gaps made up 46.5 % of missingthe NEE data record from the EddyFen station, due to unfavourable micro-meteorological conditions, instrument failures, maintenance and calibration (Jensen and Christensen 2014), but also due to the rejection of low quality flux measurements or too low u*. In 2014 a major instrument failure forced the station to stop measurements in the middle of the season. In 2010 and 2012 there were two more interruptions in the measurements (data gaps of >20 days) although the problems could be repairedsolved before the end of the season. Such prolonged gaps led to unreliable gap-filled NEE estimates. REddyProc marginal distribution sampling (MDS) algorithm tended to fill these large gaps with high peaks of respiration at noon times, coercing C uptake underestimation. For this reason, an independent AC NEE dataset (2010-2013) was tested to gap-fill EC data (Figure 3 and Figure S2; supplementary material). The $R^2$ obtained from the EC-AC correlations were always > 0.70 (2010: $R^2$= 0.80, p < 0.001; 2011: $R^2$= 0.72, p < 0.001; 2012: $R^2$= 0.80, p < 0.001; 2013: $R^2$= 0.84, p < 0.001). By using AC data, theThe number of propotionproportion of missing data gaps was reduced to 28% and it waswe found that the random uncertainty from the combination of AC and MDS algorithm decreased 5% on average. By using the u*filtering and the AC data together with EC, there was an increase of ca ~6 % in terms of C sink strength. Moreover, the propagated uncertainty in NEE never exceeded ±1.8 g C m$^{-2}$, mainly because the error related to u* filtering was low. Further, we hypothesized that different flux partitioning approaches would lead to different estimates of GPP and $R_{eco}$, however, the results suggest a relatively good agreement (Figure 4). There was a higher degree of agreement with regard to GPP ($R^2$ = 0.83) compared with $R_{eco}$ ($R^2$ = 0.30). LRC tended to calculateestimate 12 % and 15 % larger GPP and $R_{eco}$, respectively, compared to REddyProc, 12 % and 15 %, respetively.

**3.3 Inter-annual and seasonal variation of $CO_2$ ecosystem fluxes**

Overall, land-atmosphere $CO_2$ exchange measured between for the snow free periods of 2008 and -2015, omitting 2011, acted as a sink of $CO_2$, taking up -30 g C m$^{-2}$ on average (range -17 to -41 g C m$^{-2}$) (Figure 5; Table 2). The cumulative NEE

showed a characteristic pattern during the measurement period (Figure 5), with an initial loss of carbon in early spring right after snowmelt (also observed in Figure 3), followed by an intense C uptake as assimilation exceeded respiratory losses, triggered by increases in temperature, PAR and vegetation growth. This transition point matched the growing season start, when NEE switched from positive values (a net C source) to negative values (a net C sink). Eventually, the ecosystem turned again into a net C source, defining the growing season end. Even with high inter-annual variability in terms of the end of snowmelt time and growing season start/length (Table 1), the results do not show a marked meteorological effect on the NEE. The ranges in annual GPP (-182 to -316 g C m$^{-2}$) and $R_{eco}$ (144 to 279 g C m$^{-2}$) (Table 2) were >5 fold larger and more variable (CV are 3.6 and 4.1 % respectively) than for NEE (0.7 %). There was a tendency towards larger GPP and $R_{eco}$ during warmer and wetter  years (Figure S3; supplementary material), but there were no warmer and drier years during the study period. The strongest growing season $CO_2$ uptake occurred in 2012 (NEE = -74.2 g C m$^{-2}$; $GS_{length}$ = 78 days), followed by 2010 (NEE = -70.0 g C m$^{-2}$; $GS_{length}$ = 85 days) (Tables 1 and 2). A lengthening of the growing season did not increase the net carbon uptake in this study. In other words, an earlier end of the snowmelt resulting in a longer growing season length did not lead to a stronger carbon sink.

The anomalous year, 2011, constituted a relatively strong source for $CO_2$ (41 g C m$^{-2}$) and was associated with a major pest outbreak, which reduced GPP more strongly than $R_{eco}$. The larvae of the moth *Eurois occulta* data, collected from pitfall traps in the surrounding *Salix* and *Empetrum* dominated plots, showed a strong peak at the beginning of the 2011 growing season (Lund et al., 2017) coinciding with high NEE and very low GPP (Figure 4). In 2011 up to 2078 larvae were observed while other years only 14 (2008), 82 (2009), 186 (2010), 0 (2012) and 8 (2013). It is likely that the reduced primary production in the  wetland area  was a partial response to the *Eurois occulta* outbreak.

The daily aggregated NEE-GPP relationships displayed consistent linear correlation (2008-2015: $R^2$= 0.77, p < 0.001) across the assessed years (Figure 6a). The linear correlations were weaker in 2010 and 2011. A hysteresis was detected in 2010 (i.e. long growing season with higher $R_{eco}$ in autumn compared to spring), while strong C releases was observed in 2011 across June and July. The relation between GPP and $R_{eco}$, which can be understood as the degree of coupling between inputs and outputs of C, and therefore the degree of C sink strength, showed non-linear patterns (Figure 6b). The curved behaviour is likely because GPP increased more than $R_{eco}$ during early growing season, except for in 2011. Moreover, $R_{eco}$ lagged behind GPP due to (1) the vegetation green-up in the first part of the growing season and (2) the higher respiration rates due to increased biomass in the second part. The years with clearer hysteresis coincide with the years with positive temperature anomalies (i.e. 2010, 2012 and 2013) of the 2008-2015 series. It is worth mentioning the different direction (clockwise vs counter-clockwise) in the hysteresis observed these years between June, July and August. The data suggest that the clockwise 2012 hysteresis was due to greater gross C cycling (GPP and $R_{eco}$) in June and July favored by warmer conditions; while in 2010 (counter-clockwise hysteresis), the higher gross C fluxes have been measured in August with warmer and wetter conditions (Figure S4; supplementary material).

~~The strongest growing season $CO_2$ uptake occurred in 2012 (NEE =, leading to a -74.2 g C m$^{-2}$) cumulative NEE, while the weakest occurred in 2011 (NEE = -12.3 g C m$^{-2}$) it was only -12.3 g C m$^{-2}$ during the weakest growing season in 2011 (Table 2). A lengthening of the growing season (2010 was the year with longest growing season) did not increase the net carbon uptake in this study. In other words, an earlier end of the snowmelt resulting in a longer growing season length did not lead to a stronger carbon sink. The gap-filled NEE time series (Figure 3) show there was predominantly $CO_2$ uptake between 06 h and 18 h West Greenland Summer Time (WGST). The fingerprints illustrate and emphasize how variable the $GS_{start}$ and the $GS_{length}$ were across the years, but also show the difference in magnitude of the growing season regarding carbon $CO_2$ uptake.~~

3.3 Data processing and quality

The NEE gap-filling and subsequent partitioning obtained from different approaches exposed inconsistencies in performance and specific uncertainties in the seasonal C budget computation. During the eight study years, there were 46.5 % of missing NEE data from the EddyFen station due to unfavourable micro-meteorological conditions, instrument failures, maintenance and calibration (Jensen and Christensen 2014) but also due to the rejection of fluxes with deficient quality or too low u*. In 2014 a major instrument failure forced the station to stop measurements in the middle of the season. In 2010 and 2012 there were two more interruptions in the measurements (data gaps of >20 days) although the problems could be repaired before the end of the season. Such prolonged gaps led to unreliable gap-filled NEE estimates. REddyProc marginal distribution sampling (MDS) algorithm tended to fill these large gaps with high peaks of respiration at noon times, coercing C uptake underestimation. For this reason, an independent AC NEE dataset (2010-2013) was tested to gap-fill EC data (Figure S3; supplementary material). The $R^2$ obtained from the EC-AC correlations were always > 0.70 (2010: $R^2 = 0.80$, $p < 0.001$; 2011: $R^2 = 0.72$, $p < 0.001$; 2012: $R^2 = 0.80$, $p < 0.001$; 2013: $R^2 = 0.84$, $p < 0.001$). The number of gaps was reduced by 18.5% and it was found that the random uncertainty from the combination of AC and MDS algorithm decreased 5% on average. By using the u*filtering and the AC data together with EC, there was an increase of ca 6 % in terms of C sink strength. Moreover, the propagated uncertainty in NEE never exceeded ±1.8 g C m⁻², mainly because the error related to u* filtering was low. Further, we hypothesized that different flux partitioning approaches would lead to different estimates of GPP and R_eco, however, the results suggest a relatively good agreement (Figure 4). There was a higher degree of agreement with regard to GPP compared with R_eco. LRC tended to calculate larger GPP and R_eco compared to REddyProc, 12 % and 15 %, respectively.

**3.4 Environmental forcing**

The daily aggregated NEE-GPP relationships display consistent linear correlation (2008-2015: $R^2 = 0.77$, $p < 0.001$) across the assessed years (Figure 6a). The linear correlations were weaker in 2010 and 2011. A hysteresis was detected in 2010 (i.e. long growing season with higher R_eco in autumn compared to spring), while strong C releases have been observed in 2011 across June and July. The relation between GPP and R_eco, which can be understood as the degree of coupling between inputs and outputs, and therefore the residence time degree of fixed C sink strength, has shown non-linear patterns (Figure 6b). The curved behaviour is likely because GPP increased more than R_eco. Moreover, R_eco lagged behind GPP due to (1) the vegetation green-up in the first part of the growing season and (2) the higher respiration rates due to increased biomass in the second part. It is worth mentioning the high variability of C sink strength between summer months (June, July and August). The years with clearer hysteresis coincide with the years with positive temperature anomalies (i.e. 2010, 2012 and 2013) of the 2008-2015 series.

The varied importance of meteorological variables (such as PAR, $T_{air}$, VPD and Precipitation) obtained from Random Forest at different temporal scales (hourly, daily, weekly and monthly) revealed showed differences in behaviour depending on the time aggregation utilized (Figure 7). PAR dominated NEE and GPP while $T_{air}$ correlated the most with $R_{eco}$ in hourly averages, whereas $T_{air}$ became increasingly important at longer temporal aggregations for all the fluxes (Figure 7). VPD and precipitation were not found to be as important as the other variables while the use of water table depth in the analysis was discarded due to its very low impact on $CO_2$ fluxes. In general, NEE and GPP showed similar performancesdistributions of importance, reinforcing the linear relationships found between NEE and GPP (Figure 76). The standard deviation of the importance'svariables' importance (across 1000 decision trees) tended to increase at coarser time aggregations.

Changes of environmental forcing (PAR, $T_{air}$ and VPD) across diurnal, seasonal and annual time scales reveal patterns of functional responses to C fluxes. The diurnal cycle analyses on hourly data showed the changes in importance between day- and night-time (Figure 8). NEE and GPP had two predominant variables ($T_{air}$ and PAR) determining the variability at day-time. PAR was importantThere was a significant decline of $T_{air}$ importance early in the morningat dawn (06 h. WGST) and dusk (20 h. WGST), while $T_{air}$ was more important at other times. This performance indicates a threshold response to PAR, and a more continuous response to temperature, coinciding with a peak of PAR at 06 h. WGST, triggering photosynthesis

and the C uptake. T~air~ rapidly turnedretu back as a primary driver along through the day. until the range period 15-17 h. WGST, when it momentarily dropped down, again, due to PAR's influence. On the other hand, $R_{eco}$ was mainly driven by
T~air~ at both night-time and day-time. VPD and PAR barely had ahad a negligiblen impact on $CO_2$ release$R_{eco}$. The seasonal pattern importance showed PAR dominating NEE and GPP from early June to early October (Figure 8), while T~air~ and VPD became more important before and after the snow--free conditions. In terms of $CO_2$ emissionrelease ($R_{eco}$) the pattern is less clear and noisier, although T~air~ appeared to be the less limiting factor.most important variable. Finally, the annual pattern exposes a performance in line with previous results, i.e. PAR dominated NEE and GPP while $R_{eco}$ was more sensitive to variations of T~air~. Interestingly, the Random Forest analysis catchesrevealed a decrease of PAR's importance in 2011, same year exposing the sharp decrease of C sink strength.

**4 Discussion**

**4.14.1 Data processing and quality**

The NEE gap-filling and subsequent partitioning into GPP and $R_{eco}$ are needed to understand the $CO_2$ flux responses to the environmental forcing. However, these procedures expose unavoidable uncertainties in the seasonal C budget calculation (Table 2) and partial inconsistencies between approaches (Figure 4) and unavoidable uncertainties in the seasonal C budget calculation (Table 2). In this study, we used a marginal distribution sampling (MDS) gap-filling technique, an enhancement to the standard look up table (LUT). Both methods have shown a good overall performance compared to other procedures such as non-linear techniques (NLRs) or semi-parametric models (SPM), but slightly inferior to artificial neural network (ANN) (Moffat et al., 2007). However, tthe MDS gap-filling alone introduced NEE estimates out of rangehe algorithm has shown a flaw in performance across the two extensive and uninterrupted gaps in 2010 and 2012 (Figure S2; supplementary material). Quantifying the uncertainty introduced by measurement gaps is difficultcomplex (Falge et al., 2001; Moffat et al., 2007; Papale et al., 2006)(reference). One possibility would be a sensitivity analysis of time series with artificially introduced gaps (Dragomir et al., 2012; Pirk et al., 2017) (reference). But the choice of gap length and position is difficult, and would render uncertainty to the uncertainty assessment moreitself itself quite uncertainuncertain. (reference). Instead, Estimated NEE during these periods were unrealistic and led to marked NEE underestimations (i.e. lower CO2 sink strength).we used the EC prediction based on independent auto-chamber (AC) measurementsdata auto chamber (AC) observations in the gap-filling processbetween 2010 and 2013. The agreement between methodsEC and AC were always $R^2$ > 0.72 and p <0.001, and the 95% confidence interval of the predictions were reported together with the resulting uncertainties (Table 2). We understandAlthough the AC data itself incorporated a new source of uncertainty to the calculations, but we also thinkconsider this method to beis less weak sthan an unreliable gap-filling estimate. We used the AC as platform to decrease the gap length and so the total random uncertainty (Aurela et al., 2002) before the MDS algorithm can operatewas applied. AC was used together with MDS, and never was used as an independent gap-filling procedure. We understand the AC data itself incorporated new source of uncertainties to the calculations, but it is less weak than an unreliable gap-filling estimate. Overall, the AC data reduced the gap length and gap-filling uncertainties.

The NEE partitioning obtained from REddyProc and LRC suggests a relatively good agreement in model performance. The one-to-one comparison between different approaches found a better agreement with regard to GPP compared to $R_{eco}$. LRC GPP was 12 % higher than REddyProc GPP; while LRC $R_{eco}$ was 15 % higher than REddyProc $R_{eco}$. In this analysis, REddyProc produced smoother $R_{eco}$ estimates compared to the noisier GPP estimates, whereas LRC performed the other way around. This is mainly because measurement noise goes into GPP for REddyProc method, and into $R_{eco}$ for LRC method. REddyProc retrieves positive GPP values whereas LRC method results in negative $R_{eco}$ values. Both scenarios are not fully convincing, although it is not straightforward how they should be treated. By removing all positive GPP / negative

$R_{eco}$ values would risk removing only one side of the extremes. Besides night-time based (REddyProc) and day-time based (LRC) partitioning approaches, several implementations have been proposed to improve the algorithms performance. Lasslop et al. (2010) has modified the hyperbolic LRC to account for the temperature sensitivity of respiration and the VPD limitation of photosynthesis. Further, Runkle et al. (2013) proposed a time-sensitive multi-bulk flux-partitioning model, where the NEE time series was analyzed in one-week increments as the combination of a temperature-dependent $R_{eco}$ flux and a PAR-dependent flux (GPP). However, it remains uncertain under which circumstances each partitioning approach is more appropriate, especially in the boundaries between low- and high-Arctic due to the lack of dark night during polar days (whe light is not  a limiting factor for  plant growth). Since there are few methods with an unclear precision, an evaluation study on the effect of using different partitioning approaches along latitudinal gradients would be very beneficial to assess the suitability for each method.

~~The balance between the two major gross fluxes in terrestrial ecosystems, photosynthetic inputs (GPP) and respiration outputs ($R_{eco}$), has experienced larger temporal variability than NEE (CV are 3.6, 4.1 and 0.7 % for GPP, $R_{eco}$ and NEE, respectively). These results suggest that both GPP and $R_{eco}$ were strongly coupled and sensitive to meteorological conditions such as insolation and temperatures (Figure 7 and 8). Interestingly, the tendency to wetter and warmer conditions led to greater rates of C cycling associated with larger GPP and $R_{eco}$ (Figure S4, supplementary material). The mirror effect observed from the ranked cumulative GPP and $R_{eco}$ (Figure 5) also suggest that the relative insensitivity of NEE to climate could be the result of the correlated response of both GPP and $R_{eco}$. Further, this study suggests that a longer growing season does not necessarily increase the net carbon uptake (Parmentier et al., 2011), since 2012 presented stronger C sink strengths (i.e. more negative NEE) than 2010. Thus, the effects on C balance of warming from climate change are not straightforward to infer.~~

4.2 Inter-annual and seasonal variation of $CO_2$ ecosystem fluxes

The balance between the two major gross fluxes in terrestrial ecosystems, photosynthetic inputs (GPP) and respiration outputs ($R_{eco}$), displayed larger temporal variability than did NEE. These results suggest that both GPP and $R_{eco}$ were strongly coupled and sensitive to meteorological conditions such as insolation and temperature (Figure 7 and 8). Interestingly, the tendency to warmer and wetter  conditions led to greater rates of C cycling associated with larger GPP and $R_{eco}$ (Figure S3; supplementary material). This result does not entirely coincide with Peichl et al. (2014), even though they performed a similar analysis for a Swedish boreal fen. This finding points towards the complexity in the response of wetland ecosystems towards changing environmental conditions. The response is dependent on many things, such as hydrological settings, and these differ between sites. In this study, larger rates of C uptake (GPP) were linked to larger rates of C release ($R_{eco}$), with the exception of the anomalous year 2011. The relative insensitivity of NEE to meteorological conditions during the snow-free period could be the result of the correlated response of ranked cumulative GPP and $R_{eco}$ (Figure 5) (Richardson et al., 2007; Wohlfahrt et al., 2008).

This site likely receives more precipitation relative to many other tundra ecosystems, and has no permafrost, thus the NEE response to climate could be less variable. However, as Kobbefjord is located in a coastal area, it is not surprising to receive high precipitation, and other ecosystems such as coastal blanket bogs  often receive even more precipitation without a clear impact of drought effect on the NEE sensitivity (Lund et al., 2015). Furthermore, permafrost adds another layer of complexity to the C dynamics (Christensen et al., 2004; Koven et al., 2011; Schuur et al., 2015). Although some studies showed similarities of $CO_2$ fluxes in various

northern wetland ecosystems with and without permafrost (Lund et al., 2015), permafrost has strong influence on the hydrology of peatlands (Åkerman and Johansson, 2008), and therefore their topography and distribution of vegetation (Johansson et al., 2013). Especially in the context of climate warming permafrost thaw can cause large changes to the ecosystems. Further, this study agrees with Parmentier et al. (2011) and Lund et al. (2012), who suggested that a longer growing season does not necessarily increase the net carbon uptake. Here a more negative NEE indicated a stronger C sink (i.e.) in 2012 compared to 2010. Parmentier et al. (2011) hypothesized that this behavior is due to site-specific differences, such as meteorology and soil structure, and that changes in the carbon cycle with longer growing seasons will not be uniform around the Arctic. Thus, the effects of climate change on the tundra C balance of are not straightforward to infer.

NEE measured in Kobbefjord from 2008 to 2015 indicates a consistent sink of $CO_2$ (within a range of -17 to -41 g C m$^{-2}$) with exception of the year 2011 (+41 g C m$^{-2}$) (Table 2). The year 2011, associated with a major pest outbreak, reduced GPP more strongly than $R_{eco}$ (Figure 5) and Kobbefjord turned into a strong growing season C source within an episodic single growing season. The return to a substantial cumulative $CO_2$ sink rates following the extreme year of 2011 shows the ability of the ecosystem to recover from the disturbance (Lund et al., 2017). Indeed, the ecosystem not only shifted back from being a C source to a C sink, but it also changed rapidly from one year to the next. Thus we found evidencesevidence in Kobbefjord of ecosystem resilience to the meteorological variability, similar to other cases described in other northern sites (Peichl et al., 2014; Zona et al., 2014). Only a few referencesreference sites have reported similar decreases in net C uptake, but in no case as large as the one observed here. Zona et al. (2014) described an effect of delayed responses to an unusual warm summer in Alaska. Their results suggested that vascular plants, which have enhanced their physiological activity during the warmer summer, might have difficulties readapting to cooler, but not atypical, conditions, which have provoked a significant decrease of GPP and $R_{eco}$ the following year. In their study, the ecosystem returned to be a fairly strong C sink after two years, suggesting strong ecosystem resilience. Moreover, Hanis et al. 2015 have reported comparable C sink - C source variations in a Canadian fen within the growing season due to changes in the water table depth. Drier and warmer than normal conditions have triggered an increase in C source strength. Finally, during an extensive outbreak of autumn and winter moths in a subarctic birch forest in Sweden, Heliasz et al. (2011) observed a similar decrease in net sink of C (most likely due to weaker GPP) across the growing season. However, the C source strength (NEE = 40.7 g C m$^{-2}$) found in 2011 at Kobbefjord was higher compared to these other cases. To our knowledge, (Rocha and Shaver, 2011)it has not been reported such abrupt disturbance concerning C sink strength in Arctic tundra has not be previously reported excluding severely burned landscapes (Rocha and Shaver, 2011).

A combination of different factors could have led to the sharp change in C balance observed between 2010-2011, both physical and biological. The year 2010 had the highest mean annual temperature while 2011 had the lowest, 3.4 °C and -1.7 °C respectively (compared to -0.4 °C, the 2008-2015 mean annual temperature). The warmest winter-time temperature (Dec-Jan-Feb) occurred in 2010, with -2.7 °C (compared to -6.79 °C, the 2008-2015 mean wintertime temperature). These climatic conditions stimulated the thinnest (0.05 m compared to averaged 0.26 m) and short-lasting snow pack in 2010; whereas 2011 had the thickest (0.41 m compared to averaged 0.26 m) and long-lasting snow pack due to the cold summer. Consequently, 2010 experienced the longest growing season (85 days) while 2011 had the shortest (only 47 days) as well as the latest start of the growing season. The year 2010 had the warmest mean annual temperature (3.4 °C compared to the -0.4 °C mean annual temperature for 2008-2015 mean) and the warmest mean wintertime temperature (-2.7 °C compared to the -6.79 °C mean for 2008-2015 mean) (Figure 2a). These climatic conditions generated the thinnest (maximum daily snow depth of 0.3 m compared to averaged 0.9 m) (Table 1) and shortest-lasting snowpack. Consequently, 2010 had the longest growing season (85 days) and very high growing season C uptake (-70 g C/ m$^{-2}$). Increases in temperature can lead to high respiration rates during early winter (Commane et al., 2017; Zona et al., 2016), but also during the following summer (Helfter et al., 2015; Lund et al., 2012)(Helfter et al., 2015), which is related to soil temperature and snow dynamics. Further,

in Kobbefjord the year 2011 had one of the lowest mean annual temperatures and mean wintertime temperatures (-1.7 and -6.1°C respectively), which created the thickest (maximum daily snow depth of 1.4 m) and the longest-lasting snowpack, leading to the shortest growing season for the study period (only 47 days). According to Lund et al. (2012), below thick snowpack soils will be insulated from reaching low temperature, acting as lid and preventing $R_{eco}$ from being released to the atmosphere until the snowmelt period.  Finally, larvae of the noctuid moth *Eurois occulta* outbreak occurred in 2011,  overlapping the abrupt decrease of C sink strength observed. Although we cannot provide a quantification of change attributed to meteorological variations and biological disturbances, there  is evidence showing that the  moth outbreak could partially  have decreased the C sink strength in Kobbefjord. In an undisturbed scenario, the meteorological conditions in 2015, colder and dryer than the mean 2008-2015 period (Figure 2), but similar to 2011, would have stimulated similar behaviours in terms of C fluxes. However, the cumulative fluxes in 2015 (Figure 5) followed analogous patterns compared to the rest of the years. This evidence agrees with literature (Callaghan et al., 2012b; Lund et al.,  2017) on the fact that tundra systems can fluctuate in sink strength influenced by factors such as episodic disturbances or species shifts, events very difficult to predict.

~~The NEE gap filling and subsequent partitioning exposed inconsistencies in performance and specific uncertainties in the seasonal C budget computation. The uncertainties found underlays the strong challenges related to accurate gap filling and partitioning estimations. In this study, we used a marginal distribution sampling (MDS) gap filling technique, an enhancement to the standard look up table (LUT). Both methods have shown a good overall performance compared to other procedures such as non-linear techniques (NLRs) or semi-parametric models (SPM), but slightly inferior to artificial neural network (ANN) (Moffat et al., 2007). However, the algorithm has shown a flaw in performance across the two extensive and uninterrupted gaps in 2010 and 2012 (Figure S3, supplementary material). Estimated NEE during these periods were unrealistic and led to marked NEE underestimations (i.e. lower $CO_2$ sink strength).~~

[revised manuscript text omitted]

~~The analyses at different temporal scales demonstrate distinct C flux responses to different environmental forcing. The hourly variability of NEE and GPP has been found to be mostly dependent on PAR, while $R_{eco}$ was linked to $T_{air}$ primarily. In order to circumvent the potential circularity conflicts based on the use of partitioning products, we filtered day-time NEE (true GPP) and night-time NEE (true $R_{eco}$), obtaining very similar results (Table S2, supplementary material). On the other hand, the daily, weekly, and monthly C flux variability were mainly driven by $T_{air}$. These results entirely agree with Lindroth et al (2007), who recognized $T_{air}$ as a main driver for NEE seasonal trends in northern peatlands. Overall, the results indicate that environmental factors that can change rapidly (e.g. PAR) will have a high influence on short time scales. Regarding temperatures, the photosynthesis is restricted by low temperatures, so enzymatically driven processes such as carbon fixation are more sensitive to low temperatures than the light-driven biophysical reactions (Chapin et al., 2011).~~

~~The changes of environmental forcing across diurnal, seasonal and annual time scales unmask patterns of functional responses to C fluxes. Interestingly the Random Forest analyses revealed a strong diurnal pattern with a marked contribution of $T_{air}$ to variations in NEE and GPP (both at night-time and between 08-18 h WGST) while $T_{air}$ was more important involving $R_{eco}$. It is also interesting to see how PAR increased importance at 08 h and 20 h WGST, coinciding 
[revised manuscript text omitted]

| | 2008 | 2009 | 2010 | 2011 | 2012 | 2013 | 2014 | 2015 | of the |
|---|---|---|---|---|---|---|---|---|---|
| First measurement (DOY) | 157 | 135 | 124 | 135 | 158 | 149 | 150 | 177 | auto- |
| Last measurement (DOY) | 303 | 304 | 282 | 287 | 305 | 295 | 209* | 294 | chamb |
| Missing data (%) | 57.6 | 42.3 | 28.6 | 35.4 | 32.3 | 29.8 | 44.9* | 40.0 | er, |
| NEE in measuring period (g C m$^{-2}$) | -41.3 | -16.9 | -24.4 | 40.7 | -37.0 | -28.1 | -28.7* | -31.5 | rando |
| | ±1.4 | ±1.4 | ±1.9 | ±1.3 | ±1.8 | ±1.7 | ±1.1 | ±1.6 | m and |
| NEE in growing season (g C m$^{-2}$) | -62.3 | -45.9 | -70.0 | -16.2 | -74.2 | -69.7 | -35.3* | -55.8 | u* |
| Maximum daily uptake (DOY) | 195 | 205 | 182 | 230 | 204 | 220 | 192* | 199 | filterin |
| Maximum uptake (µmols m$^{-2}$ s$^{-1}$) | -2.4 | -1.7 | -3.0 | -1.4 | -2.8 | -2.5 | -1.9* | -2.3 | g |
| Estimated GPP (g C m$^{-2}$) | -185.5 | -181.8 | -266.1 | -130.6 | -316.2 | -230.7 | -106.8* | -206.0 | uncerta |
| | ±1.4 | ±1.4 | ±1.9 | ±1.3 | ±1.9 | ±1.7 | ±1.1 | ±1.6 | inties, |
| Estimated R$_{eco}$ (g C m$^{-2}$) | 144.2 | 164.9 | 241.6 | 171.3 | 279.2 | 202.6 | 78.1* | 174.6 | * |
| | ±1.3 | ±1.3 | ±1.8 | ±1.2 | ±1.8 | ±1.7 | ±1.1 | ±1.5 | |

965

970

where applicable: ± sum of the auto-chamber, random and u* filtering uncertainties, * incomplete growing season dataset.

975